# Probabilistic and Statistical Techniques to Study the Impact of Localized Corrosion Defects in Oil and Gas Pipelines: A Review

**Julio César Velázquez [1,\*], Enrique Hernández-Sánchez [2], Gerardo Terán [3], Selene Capula-Colindres [3], Manuela Diaz-Cruz [4] and Arturo Cervantes-Tobón [4]**

[1] Departamento de Ingeniería Química Industrial, ESIQIE, Instituto Politécnico Nacional, UPALM Edif. 7, Zacatenco, Mexico City 07738, Mexico

[2] Departamento de Bioingeniería, Instituto Politécnico Nacional, UPIBI, Avenida Acueducto s/n Barrio La Laguna Ticomán, Mexico City 07340, Mexico; enriquehs266@yahoo.com.mx

[3] Departamento de Metalurgia, IPN-CECYT 2, Av. Casa de La Moneda 133, Lomas de Sotelo, México City 11200, Mexico; gteranm@ipn.mx (G.T.); sicapula@ipn.mx (S.C.-C.)

[4] Departamento de Metalurgia y Materiales, ESIQIE, Instituto Politécnico Nacional S/N, UPALM Edif. 7, Zacatenco, Mexico City 07738, Mexico; mdiazc@ipn.mx (M.D.-C.); acervantest@ipn.mx (A.C.-T.)

\* Correspondence: jvelazqueza@ipn.mx

**Abstract:** Corrosion is a major cause of the loss of hermeticity in oil and gas pipelines. Corrosion defects affect the remaining life of in-service pipelines and can lead to failures, ruptures, hydrocarbon leakage, product loss, interruptions, environmental damage, economic losses, or, in the worst cases, fatalities. The existence of localized corrosion defects is a significant issue in pipeline integrity analysis, mainly because these structures are commonly buried and cover large extensions, amounting to hundreds or even thousands of miles; thus, it is difficult to size and locate all minor but possibly deep defects. Consequently, probabilistic and statistical modeling methods have been widely used to assess the integrity of corroded pipelines. Statistical modeling methods used to estimate the remaining life of the pipeline have focused on three main aspects: applications to estimate the defect depths and rates of corrosion, Bayesian applications in pipeline integrity to update the probability distribution for corrosion defects (depth, length, and spatial distribution), and pipeline reliability estimations. This paper reviews several methods proposed in the literature for these issues as well as their applications in real life. In addition, some of the present and future challenges related to preventing corrosion in the oil and gas pipeline industry are described.

**Keywords:** corrosion defect; pipeline; oil and gas; probability and statistics; modeling; carbon steel

## 1. Introduction

A shift in the global energy system is desirable to alleviate the negative impacts of global climate change, taking advantage of the cost reduction that has occurred in renewable energies (between 40 and 50%) [1]. This shift could help avoid the deterioration of the quality of life of millions of people. However, fossil fuels are indispensable for economic growth in many countries. Some studies have indicated that the oil demand will probably peak in the second part of the next decade (2030–2040) but will be highly demanded in developing countries for more time [2]. Therefore, it can be concluded that crude oil, its derivatives, and natural gas will remain important in the energetic matrix, especially in emerging countries.

Pipelines are the most lucrative method of transporting gas and liquid hydrocarbons [3,4]. Thus, it is important to generate methodologies and technologies that reduce leaks and ruptures in these structures, keeping them in safe operating conditions. There is approximately 3.5 million km of oil and gas pipelines worldwide [5,6]. These pipelines

usually fail because of corrosion deterioration, defects in welding, dents, third-party damage, cracking, and other reasons. The authors of [6] showed that in Canada, the United Kingdom, Europe, and the United States, a major reason for pipeline failures is corrosion damage. The European Gas Pipeline Incident Data Group has recognized that the damage caused by corrosion mechanisms has increased in recent years, from 16.7 to 26.6% [7]. Because most of these leaks could be caused by a localized corrosion mechanism [8], it is of significance to use all the available techniques to study the impact of localized corrosion defects in pipeline operations. For instance, in British Columbia, Canada, approximately 1 million cubic meters of natural gas leaked owing to pipeline damage in 2012 [8]. In addition to the threat that corrosion is for structural integrity, the costs derived from this have not yet been particularly studied for the pipeline industry worldwide; however, according to a report published by NACE in 2016, the estimated global cost of corrosion is approximately USD 2.5 trillion [9].

Probability and statistics have been a significant tool to study the impact of pipeline and vessel corrosion since the 1930s [10], and the first study that incorporated the concept of "corrosion probability" was conducted in 1933 [10]. Two decades later, other important studies that applied statistical concepts were performed by Romanoff in 1957 [11], applying the concept of linear regression (the process used by Romanoff [11] is detailed later in this paper in Section 3), and Aziz in 1956 [12], using probability density functions to study the pitting corrosion phenomenon on aluminum. A major reason for the use of probability and statistics is the random nature of the corrosion phenomenon owing to the lack of homogeneity of the metals, differences in the chemical composition of the environment, changes in the temperature, variations in the pipeline direction, drop in pressure, intermittent cathodic protection, and coating disbondment. Usually, in the petroleum industry, it is necessary to estimate the remaining life of pipelines, and a solution that apparently sounds logical would be to search previous studies for the corrosion rate of this material in contact with a specific environment to perform this estimation. However, it is difficult to extrapolate this bibliographic corrosion rate to a service pipeline because of the aforementioned reasons and the different corrosion mechanisms that can be undergone [13]. Consequently, it is essential for corrosion specialists and technologists to study in detail the corrosion probability, which is usually of practical significance.

Several pipeline corrosion studies have been conducted using probability and statistics as a tool to manage structural integrity, thereby reducing the risk of leaks and ruptures. In this context, the present review summarizes the most used and recognized statistical techniques applied in oil and gas pipelines and some exemplifications of these. In addition, the basics for these statistical techniques are explained in each section with the purpose of simplifying the information search for future specialists.

To describe some applications of statistics and probability techniques in oil and gas pipelines that undergo damage from localized corrosion defects, this study is divided into the following sections:

- Electrochemical background for the statistical modeling of localized corrosion defects. The details, background, and electrochemical concepts needed to model localized corrosion defects are explained. Similarly, the chemical and physical factors that influence the growth of corrosion defects are described. In addition, the applications of statistics are explained to better understand the electrochemical nature of the localized corrosion phenomenon.
- Estimation of corrosion defect depths and corrosion rates. Quantification of the uncertainty of the localized corrosion defect depth and rate assist in the estimation of the thickness of the wall of the pipeline with accuracy and precision and, thus, the remaining life of these structures [14–16]. Consequently, in this section, the use of different statistical techniques as tools for estimation is explained.
- Bayesian applications in pipeline integrity to update the probability distributions of corrosion defect characteristics (depth, length, and spatial distribution). This statistical technique can be applied to estimate the depth of the corrosion defect, corrosion

rate, and the sample size required to estimate the depth of corrosion defect and other damage caused by different corrosion mechanisms [17–20]. In this section, an application that used Bayesian inference is detailed.

- Pipeline reliability estimations. Reliability analysis has become a cornerstone in pipeline integrity management to mitigate the threats provoked by different corrosion mechanisms. The reliability of corroded pipelines is usually assessed by probabilistic tools that should consider the unavoidable uncertainties associated with the sizing of corrosion defects, the pipe manufacturing process that influences the material mechanical characteristics, pipe dimensioning, and working conditions of these pipelines [16,21–23]. In this section, several examples of pipeline reliability estimation are shown to broadcast the scope of probability and statistics in pipeline integrity management.

- Future challenges for the application of probability and statistics in corroded oil and gas pipelines. The challenge of the application of probability and statistical techniques is discussed to motivate their use or at least gradually reduce the risk caused by corrosion defects in pipelines that transport hydrocarbons.

## 2. Electrochemical Background for Statistical Modeling of Localized Corrosion Defects

Oil and gas pipelines that are manufactured from low-carbon steel suffer localized corrosion attacks that can provoke leaks [6,8,24,25]. Usually, this localized corrosion phenomenon is mainly caused by pitting corrosion and coating defects or coating disbandment in buried pipes [26]. In steels, pitting corrosion occurs when the surface areas of the anodic and cathodic sites differ [27]. This difference could be caused by differential aeration cells or dissimilar soils in underground pipelines [28]. The pitting corrosion rates in underground pipelines were frequently higher than those in other areas. In pitting corrosion, the amount of metal lost is small because the phenomenon originates from a relatively small area, causing insidious damage. Cavities caused by pitting corrosion can show different shapes, as shown in Figure 1 in the studies conducted by Bhandari et al. [29]. However, depth has a more significant influence on failure pressure pipelines [22]; therefore, estimating the future corrosion defect depth rate is important to plan future repairs and reduce the risk in operating pipelines. Pitting corrosion is an electrochemical process that involves four stages.

The first stage is passive film breakdown [30] (passive film refers to the formation of an ultrathin film of corrosion products (usually known as rust for iron or carbon steel) on the surface of the material that works as an obstacle to further oxidation [30]). Agar and Hoar began to investigate the passive film-breaking theory in the 1940s [31]. Pitting corrosion occurs when there is a breakdown of the surface films exposed to corrosive environments. This breakdown can provide sites for pit nucleation; therefore, these breakdown sites are more vulnerable to cavities. The polarization curve, illustrated in Figure 1, allows the estimation of the susceptibility of the material to pitting corrosion. In this polarization curve, the pitting potential ($E_{pit}$), that is, the potential at which pits could be formed, can be determined. It is the potential at which the salt of an ion in any solution is in equilibrium with the metal oxide [32]. A greater $E_{pit}$ for a metal in a specific environment indicates higher resistance to pitting [33]. Similarly, the repassivation potential ($E_{rp}$) occurs during the repetition of a previous passivation process. This electrochemical process is used to prevent the corrosion of metals by reducing chemical reactivity. Repassivation reiterates the process of passivation after a period because the protective film has worn away [34]. Figure 1 also illustrates the localization of the repassivation potential in the polarization curve. In any situation where the potential is located between $E_{pit}$ and $E_{rp}$, pitting corrosion can occur [33].

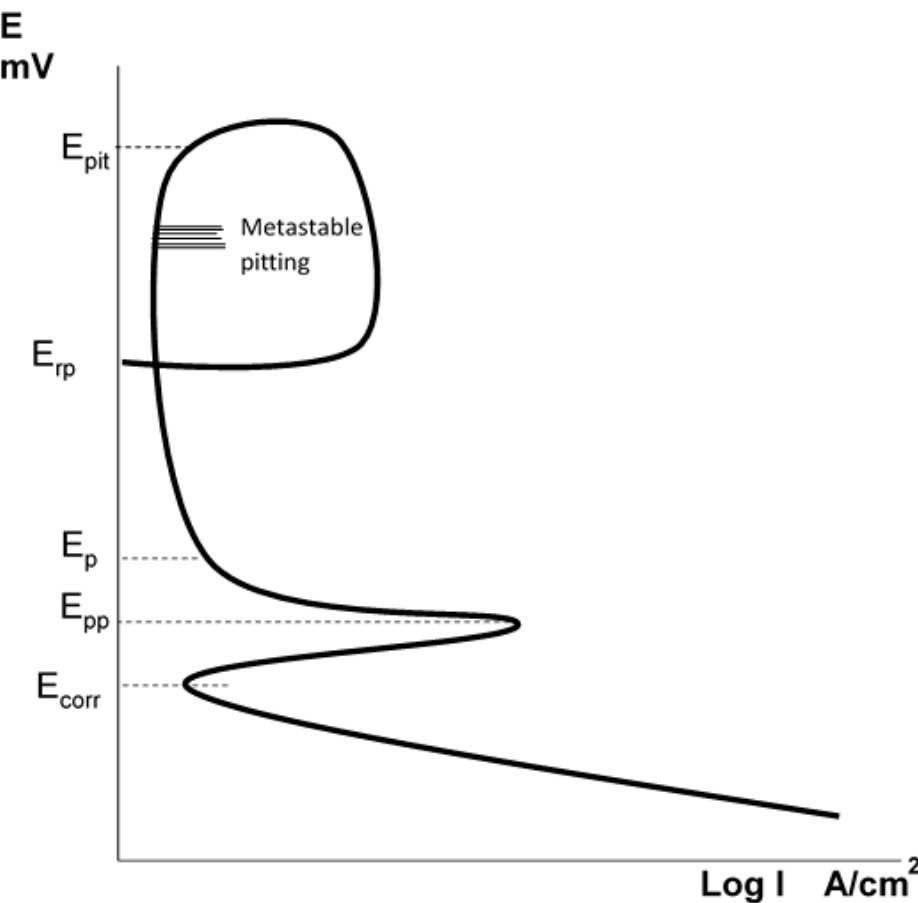

**Figure 1.** Schematic polarization curve: $E_{pit}$ pitting potential; $E_{rp}$ repassivation potential; $E_{pp}$ primary passive potential; $E_p$ passive potential; $E_{corr}$ corrosion potential.

The second stage is pit initiation. Pit initiation is usually influenced by material defects that could result from the manufacturing process, such as low quality of raw materials, lack of control variables, and installation problems. Some factors provoke the initiation of pits, such as damage to the oxide film, environmental characteristics, or film discontinuities [35].

The third stage is metastable pitting. These pits initiate and grow for a period before repassivation and dying [36]. Pits that stop growing and repassivate can be recognized as metastable. Electrochemical studies on localized corrosion processes indicate that dissolutions of MnS inclusions can play an important role in metastable pitting [37]. Metastable pits, such as the early growth of initiated pits, continue to live to become stable growing pits [38].

The fourth stage is pit propagation and growth. Pit propagation and growth is the stage where the growth of some of the initiated pits occurs and their rate of growth can increase or stabilize [37]. Some conditions must be met for this stage to be achieved; for example, the pitting potential must exceed the repassivation potential, an aggressive ion must be present (especially halogen ions), and a notorious breakdown of the protective film must occur [39].

Conversely, disbonded coating on carbon steel pipelines is another common cause that provokes localized corrosion. Disbondment is a common failure mode in pipeline coatings [40]. Disbonding is created by a non-adequate coating process, naturally existing crevices in a certain coating, differential soil stresses, or an overpotential in a cathodic protection system. These flaws in coated pipelines are called "holidays" and are responsible for the appearance of localized corrosion defects. Localized corrosion defects (by pitting or by coating disbondment) are always influenced by physical heterogeneities of the material and the environment or by changes in the chemical composition of the environment [33].

It is well known that passive film breakdown can occur in the presence of active anionic species. One of these species is chlorides. There is a multitude of corrosion reactions; however, the main and most common reactions are as follows [41]:

$$\text{Oxidation: Fe} \rightarrow \text{Fe}^{+2} + 2e^- \tag{1}$$

$$\text{Fe}^{2+} \rightarrow \text{Fe}^{3+} + e^- \tag{2}$$

$$\text{Reduction: } 2H^+ + 2e^- \rightarrow H_2 \text{ (acidic solutions)} \tag{3}$$

$$O_2 + 4H^+ + 4e^- \rightarrow 2H_2O \tag{4}$$

$$O_2 + 2H_2O + 4e^- \rightarrow 4OH^- \text{ (neutral or basic solutions)} \tag{5}$$

Many factors influence the kinetics of localized corrosion defect growth. The main factors are as follows:

- Temperature. The higher the temperature, the higher the corrosion rate because of the resulting accelerated electrochemical reactions. Nonetheless, under some conditions, the effect of temperature on protective layer formation is multivariate because with a significant increase in temperature, protective films are formed faster and reduce the deterioration [42].
- pH. Frequently, the acidity or basicity of the environment has been recognized as the variable that exerts more influence on pitting corrosion deterioration [12,43]. Because the effect of pH on the corrosion rate is not completely understood and depends on the environment, it is not feasible to conclude that the relationship with the corrosion rate is always inversely proportional. In some studies, it was observed that the value of the breakdown film potential ($E_b$) [44] (in electrochemical techniques, breakdown potential is the surface potential at which the surface's passive film breaks down [42]) is almost flat within a large range of pH values [45].
- Chemical composition. Some ions that encourage localized corrosion deterioration, such as halides (mainly chlorides), can form salts at low pH at the bottom of the pit [42,46]. However, other ions, such as bicarbonate, carbonate, and sulfate, discourage the growth of localized corrosion defects [42,46].
- Fluid velocity. For metals such as steel, there is a critical velocity beyond which the corrosion rate is high. It is important to remember that when the flow velocity increases, the protective layers detach from the surface, indicating that the corrosion rate increases as the velocity increases [42,47,48].
- Biological factors. Microbiologically influenced corrosion has been reported to cause approximately 40% of all internal corrosion incidents in oil pipelines [49,50]. Sulfate-reducing bacteria are recognized as the major bacteria that cause corrosion. These bacteria are anaerobic and can degrade organic compounds to produce sulfides [51].
- Metallurgical factors. Some characteristics of steel can influence the growth of localized corrosion defects, such as inclusion density, alloy composition, surface finish, grain size, and grain boundary [29].
- Dissolved gases in the environment. The fluids transmitted by oil and gas pipelines contain dissolved gases such as carbon dioxide ($CO_2$), oxygen ($O_2$), and hydrogen sulfide ($H_2S$). Carbon dioxide reacts with water and leads to the formation of carbonic acid ($H_2CO_3$), which decreases the pH of the fluid, making it more aggressive and deteriorating the metal surface. Nevertheless, carbonic acid dissociates in hydrogen and bicarbonate ions, and the bicarbonate ion can reduce the corrosion rate. Therefore, the influence of carbonic acid on the corrosion rate depends on the amount present and the interaction with other ions and the physical variables involved [42,46]. Oxygen is also present in the hydrocarbons that are transmitted by pipelines and is also present in soils surrounding these structures. This means oxygen has an influence on both external and internal corrosion in pipelines because it increases corrosion rates [42]. The corrosion rate of local anodes depends on the cathode reaction; therefore,

depolarization is faster with an increase in oxygen concentration at the cathode [45]. $H_2S$ is also found in oil and gas. This gas is approximately 3 times more soluble than $CO_2$. $H_2S$ can also reduce pH, such as $CO_2$, resulting in a higher corrosion rate [41].

Other factors can influence the localized corrosion rate, such as pollutants, fouling, atmospheric effects, or capacity of an alloy to resist pitting (pitting resistance equivalent); however, these factors can be considered to be dependent on the aforementioned main variables.

Because a considerable number of variables are involved in the growth of defects caused by localized corrosion, some of which do not have a clear type of influence, and because there are several stages in the growth of these defects, it is common to use statistical and probabilistic techniques to study the phenomenon. As Aziz mentioned in [12], the randomness of the pitting corrosion phenomenon is because of the factors mentioned above and other minor factors (microscopic faults of the metal, weak spots in the oxide film, or ion diffusion in the electrolyte) acting in a random fashion and producing erratic but predictable results.

The random nature of localized corrosion using electrochemical techniques has been studied. The pitting potential is usually measured to determine the susceptibility of any metal to pitting corrosion. If the experiment is performed M times, it is feasible to obtain a potential pitting distribution with $Epit_1$, $Epit_2$, ... , $Epit_i$, ... , $Epit_m$, which allows the determination of the distribution function of the probability of the pitting potential. For example, Shibata and Takeyama [52] proposed the use of a multichannel pitting corrosion device to measure the pitting potential and induction time (pit initiation time) for 12 specimens at the same time (Figure 2). These authors used stainless steel samples immersed in NaCl solution to perform the experiments. They observed that the pitting potential under this condition can be fitted to a normal distribution, and the induction time histogram has a right-skewed characteristic. Shibata and Takeyama [52], using the information obtained from this experiment and assuming the pit generation process has the Markov property, observed that the linear dependence of the pit generation rate on the potentials suggest that it has more influence on pitting corrosion and the breakdown of passive film than electrochemical reactions. In contrast, Gabrielli et al. studied the probability distribution of the induction times with 100 samples [53]. They fitted the observed data to a lognormal distribution (right-skewed distribution) with excellent accuracy [53]. The prepitting stage can be modeled by a birth and death process [53], using the following differential equation:

$$\frac{dp(t,n)}{dt} = -(\lambda + n\mu)p(t,n) + (n+1)\mu p(t,n+1) + \lambda p(t,n-1) \tag{6}$$

where $\lambda$ is the birth rate of the pits, $\mu$ is the death rate, and n is the number of pits generated at time t.

Therefore, the use of both statistics and electrochemical techniques helps to better understand the nature of the corrosion mechanism; however, the direct application of these in oil and gas pipelines is challenging because the sample is not under control and the metal structure is in contact with numerous variables added to those already indicated.

In recent decades, newer electrochemical techniques, such as electrochemical impedance spectroscopy (EIS) or electrochemical noise, have been used to study the different corrosion mechanisms. However, statistical approaches that consider the information generated from these studies are missing. This issue could be taken into account for future studies.

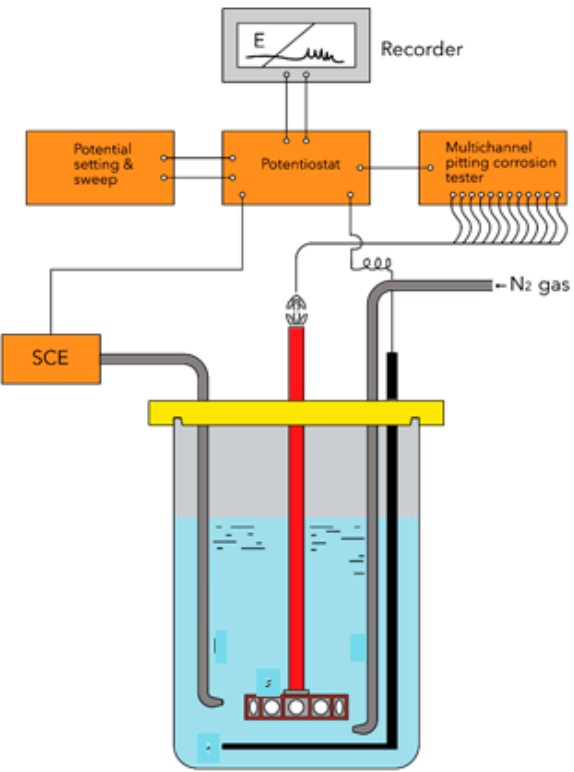

**Figure 2.** Illustration of the multichannel pitting corrosion testing device. Adapted and turned into color according to the scheme given in Figure 1 of reprinted with permission from ref. [52]. Copyright 2022 CORROSION.

## 3. Applications of Probabilistic and Statistical Methods to Approximate Localized Corrosion Defect Depth and Rate in Pipelines

To study localized corrosion using electrochemical techniques, it is necessary to control the sample size, chemical and physical characteristics of the environment, and potential supply. This makes it impossible to use electrochemical techniques in steel pipelines, which are usually buried in different soils. In addition, the pipelines suffer repairs, and some sections are replaced with new pipes. This alters the homogeneity of pipeline metallurgy. Accordingly, statistics and probabilistic methods have been used to model and estimate the localized corrosion defect depth and its corrosion rate. The recognition of the random nature of the corrosion started in the 1930s; however, detailed studies on corrosion damage using probabilistic and statistical methods only began after three pioneering studies that were published in the 1950s. These three studies, which were led by Lewis, Romanoff, and Aziz [11,12,54], observed that the corrosion rate has a random nature, recognizing that a considerable number of variables always exist that cannot be controlled or their control would be uneconomical. Lewis focused on the explanation of the sample size and estimation of the mean using the normal distribution.

Conversely, in [11], several studies on the corrosion of buried metallic samples are condensed. Romanoff proposed a model of pit growth using the following power law:

$$y_{max}(t) = At^b \tag{7}$$

where $y_{max}$ is the depth of the deepest pit at time $t$ and $A$ and $b$ are the parameters to be determined. Romanoff used the characteristics of the deepest pit in a sample. In studies on mechanical integrity, it is recognized that the deepest pit is the most dangerous defect because this tiny defect, which is very difficult to detect, can provoke leaks. From this power law (Equation (7)) and taking logarithms, it is possible to obtain a straight-line equation to obtain the values of $A$ and $b$. After this process, it is possible to plot the power

law and estimate the maximum pitting depth in the future. Figure 3 shows the behavior of the power law (Equation (7)) for different soils in the United States, using the information described in Table 18 in [11]. From Figure 3, it can be observed that parameter *A* (mils) works as a scale parameter, and *b* works as a shape parameter.

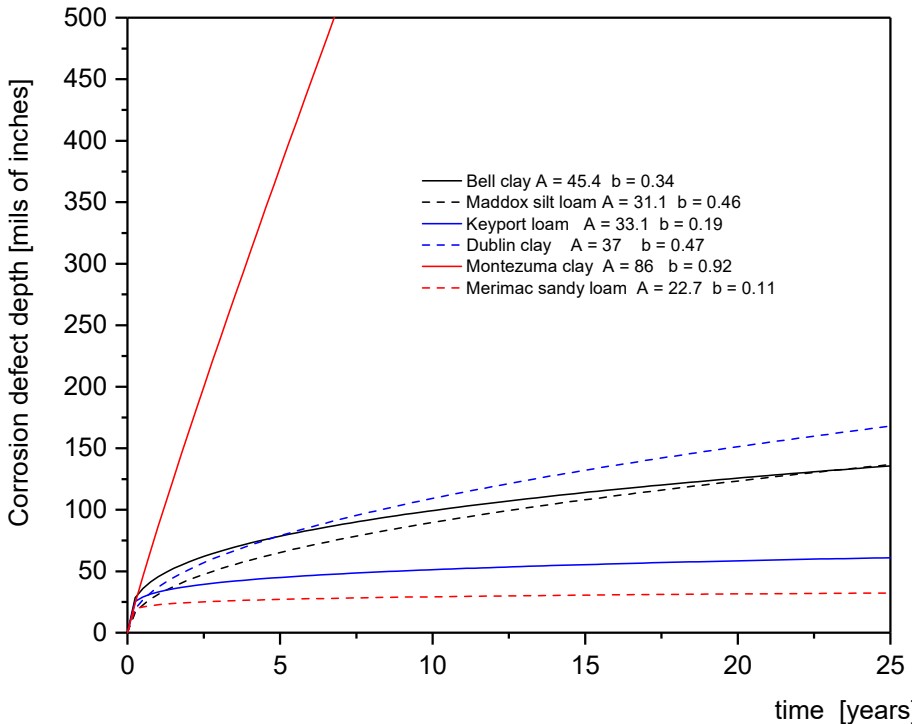

**Figure 3.** Evolution of the corrosion defect depth according to the information presented in Table 18 in the work titled *Underground Corrosion* by Romanoff adapted from [11].

Another classic important study was done by Aziz [12], where he focused on studies of pitting corrosion phenomena using a probability interpretation because of the lack of reproducibility in some previous experiments. Aziz observed that the maximum pit depth can be studied using the statistical theory of extreme values, fitting the histogram of maximum pit depths generated in an aluminum alloy immersed in tap water to an asymptotic distribution. Gumbel distribution was chosen because it can represent a skewed distribution with a long tail, reproducing the maximum histogram of the pit depth. The Gumbel distribution function (probability density function (PDF)) is represented by the following mathematical expression:

$$f(x) = \frac{1}{\ddot{\sigma}} e^{-\frac{y-\ddot{\mu}}{\ddot{\sigma}} - e^{-\frac{y-\ddot{\mu}}{\ddot{\sigma}}}} \tag{8}$$

where *y* is a random variable (in this case, maximum pit depth), $\ddot{\mu}$ is the location parameter, and $\ddot{\sigma}$ is a scale parameter. Gumbel distribution is a skewed distribution; it is shown in Figure 4 for different values (the values of location and scale parameter were taken from [55], where block maxima and peak over threshold approaches to extremes have been applied to pitting corrosion data from the immersion test for line pipe steel) of $\ddot{\mu}$ and $\ddot{\sigma}$. The maximum pit depth exhibits a linear behavior in the logarithm of the exposed area. Thus, it is possible to extrapolate pitting data obtained from small samples to large-scale installations.

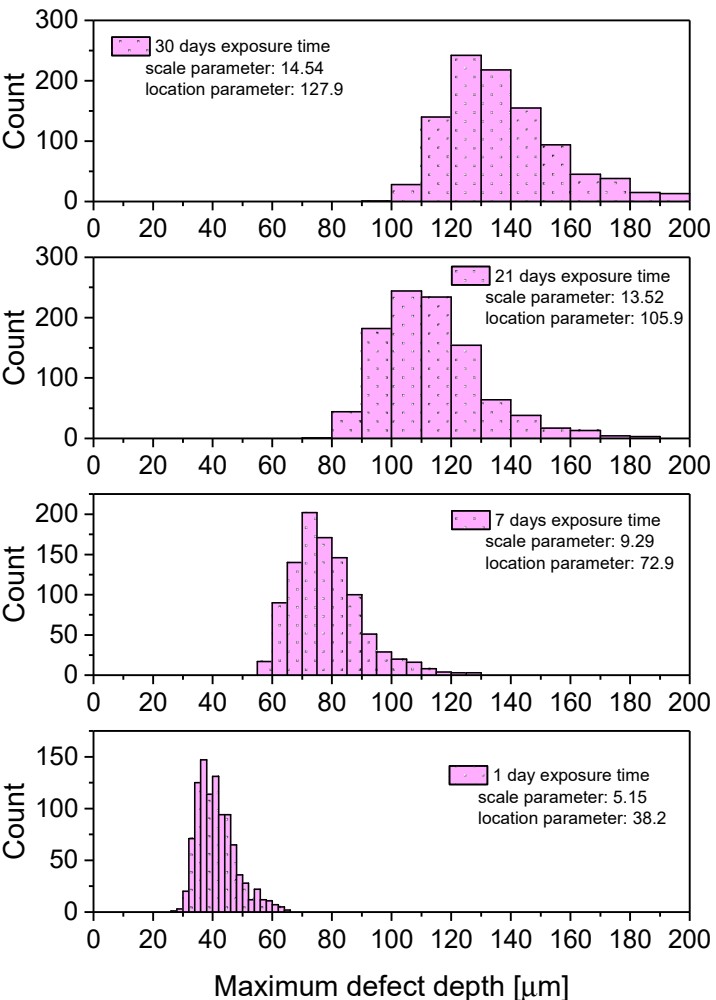

**Figure 4.** Histograms generated using Monte Carlo simulations for different values of location and scale parameters of Gumbel PDF. The parameters values are taken from Table 4 data from [55].

*Advances in Regression Models*

Parameters $A$ and $b$ from the power-law model used by Romanoff [11] are influenced by the soil's physical and chemical characteristics. From this viewpoint, [56] proposed the relation of these parameters ($A$ and $b$) to soil properties and determined a practical approach to estimate the pitting defect depth in buried samples of carbon steels and ferrous metals obtained by the National Bureau of Standards (NBS). They related parameter $A$ with the soil's pH using linear regression analysis and found two mathematical expressions: one for acidic soils and the other for alkaline soils; these are represented in Equations (9) and (10), respectively.

$$A_a = 5.74(9.9 - \text{pH}) \tag{9}$$

$$A_b = 5.05(2\text{pH} - 10.3) \tag{10}$$

Parameter $b$ was related to both the clay content (CL) and moisture content ($\theta$) of the soil. The linear model that allows the computation of this parameter $b$ is shown in Equation (11).

$$b = a_1\theta + a_2\text{CL} + a_3 \tag{11}$$

After the study done by Mughabghab and Sullivan [56], it can be inferred that there is a correlation between the maximum pit depth and soil characteristics. Kajiyama and Koyama [57] investigated the correlation between soil characteristics and the maximum pit depth in a buried pipe and the correlation among the same soil characteristics. These

authors observed a considerable correlation between pH, soil specific gravity, and pipe-to-soil potential with the maximum pit depth. Kajiyama and Koyama showed that several soil parameters are worth studying and should be included in a model. In the present century, and because the corrosion rate of steel and iron pipes depends on the corrosiveness of soils, Katano et al. [58] proposed using a regression model that includes 20 variables. This model is not linear and is represented by (12):

$$y_{max}(t) = Exp(\alpha_0 + \sum_{j=1}^{p} \alpha_j x_j) t^b \tag{12}$$

where $\alpha$ and $b$ are regression coefficients, and the methodology to determine these parameters was developed using the maximum likelihood estimator. It was observed that soil type, preparation history of the land, soil resistivity, pH, redox potential, and sulfide content were the variables that significantly influenced the pit growth.

Race et al. [59] proposed another model in which a model to estimate the corrosion rate in pipelines was developed. This model considers the coating type, coating condition, coating age, cathodic protection effectiveness (availability and maintenance), soil type, specific corrosion rate for specific soils, and inspection factor. Race's model, a scoring model, is not detailed in this study because it is not a regression model, but it is the first model that incorporates the two most recognized and used methods to reduce pipeline corrosion damage: coating and cathodic protection. This scoring model includes the effects of the tool technology used to inspect the pipeline.

Papavinasam et al. [42] developed a statistical model that is worth analyzing. They built a model that incorporates pipeline construction parameters (pipe diameter, pipe wall thickness, and pipe inclination) and 11 operational parameters (production rates of oil, water, gas, solid, temperature, total pressure, partial pressures of hydrogen sulfide, carbon dioxide, concentrations of sulfate, bicarbonate, and chloride). Hence, according to Papavinasam et al. [42], the pitting corrosion rate using the operational parameters of oil and gas pipelines can be calculated using Equation (13):

$$\begin{aligned} &\{[\sum(-0.33\theta + 55) + (0.51W + 12.13) + (0.19W_{ss} + 64) + (50 + 25R_{solid}) + \\ &(0.57T + 20) + (-0.081P_{tot} + 88) + (-0.54P_{H_2S} + 67) + \left(-0.013C_{sulfate} + \right. \\ &\left. 57\right) + \left(-0.63O_{CO_2} + 74\right) + (-0.014C_{bicarbon} + 81) + (0.0007C_{chloride} + \\ &\left. 9.2\right) + CR_{gen}] / 12\} \text{x} 1/t \end{aligned} \tag{13}$$

One advantage of this model is that it is applicable to both sweet (corrosion primarily caused by dissolved $CO_2$ is usually called "sweet" corrosion [60]) and sour conditions (corrosion caused by the combined presence of dissolved $CO_2$ and hydrogen sulfide ($H_2S$) or only $H_2S$ is referred to as "sour" corrosion [60]) [60]. Papavinasam's model was validated using data obtained from seven operating fields, making it highly applicable.

The damage caused by corrosion in pipelines can be subjected to both external and internal factors, and because of the difference in environments, a particular model should be used for each. For both external and internal localized corrosion, it is feasible to use a power law, as shown in Equation (12). In this sense, Velázquez et al. [14,46] used a nonlinear regression technique with a combinatory methodology to find a model that can estimate the external corrosion defect depth [14] and the internal corrosion defect depth [46]. The first step to model the external corrosion in buried pipelines is to establish which variables need to be included because of soil corrosion. The variables considered by Velázquez et al. in 2009 [14] to model the corrosion defect growth in buried oil and gas pipelines are redox potential, pH, pipe-to-soil potential, soil resistivity, water content, soil density, chloride content, bicarbonate content, sulfate content, and coating type. Coatings were categorized into five categories according to their type: fusion-bonded epoxy, coal tar, wrapped polyolefin tape, asphalt enamel, and non-coated or bare pipe. All the data and information of these variables, including the maximum pit depth, were collected by

field studies performed previously in oil and gas pipelines located in Mexico (Velázquez et al. shared all data and information that they used to developed their statistical model in [61]). The coating categorization and its scores were the same as those proposed by Race et al. [49]. After defining the variables to study, Velázquez et al. [14] established the influence of the predictor variables on the parameters of the well-known power-law model (*A* and *b*), expressed as linear combinations of the soil and pipe variables:

$$y_{max}(t) = A(t - t_{sd})^b \tag{14}$$

$$y_{max}(t) = (\alpha_0 + \sum_{j=1}^{p} \alpha_j x_j)(t - t_{sd})^{(\beta_0 + \sum_{k=1}^{q} \beta_k x_k)} \tag{15}$$

where $x_j$ and $x_k$ symbolize the *j*-th and *k*-th predictor variables, $\alpha_j$ and $\beta_k$ are the regression coefficients for this predictor, and $t_{sd}$ is the pit initiation time.

Velázquez et al. [14,61] observed different types of soils in the zone where the pipeline inspections were performed. The most frequent were clay, clay loam, and sandy clay loam. For each soil category, a regression analysis was performed for each of the 1024 potential combinations for the distribution of predictor variables between *A* and *b*. This process was also performed for all the soils found. The best model obtained was chosen based on the highest coefficient of determination ($R^2$). The model has the following parameters:

$$A = \alpha_0 + \alpha_1 rp + \alpha_2 pH + \alpha_3 re + \alpha_4 cc + \alpha_5 bc + \alpha_6 sc \tag{16}$$

$$b = \beta_0 + \beta_1 pp + \beta_2 wc + \beta_3 bd + \beta_4 ct \tag{17}$$

where $rp$ is the redox potential, $pH$ is the soil pH, $re$ is the resistivity, $cc$ is the chloride content, $bc$ is the bicarbonate content, $sc$ is the sulfate content, $pp$ is the pipe-to-soil potential, $wc$ is the water content, $bd$ is the soil bulk density, and $ct$ is the coating type. Parameters $\alpha_i$ and $\beta_i$ are regression parameters, and their values are listed in Table 7 in [14].

In addition to estimating the maximum corrosion defect depth in a buried pipeline, the model developed by Velázquez et al. [14] can be used to perform a sensitivity analysis. In this analysis, it was observed that the variables that exert more influence on corrosion defect growth are pH, pipe-to-soil potential, pipeline coating type, soil bulk density, water content, and chloride content, in that order. Table 8 of [14] shows the average values for the parameter of the power law (*A* and *b*), and for the sake of illustration, Figure 5 shows the schematic of the pitting growth rate at typical conditions, showing the aggressiveness of the clay soils.

The same methodology used to develop a model to estimate the growth of an external defect in buried pipelines proposed by Velázquez et al. [14] was applied to model the internal corrosion defect growth in pipelines that transmit produced waters. This is shown in another study [46] using immersion tests in the laboratory of pipeline steel samples under a synthetic oilfield-produced water environment. The independent variables included modeling of the localized corrosion growth as follows: pH, redox potential ($orp$), conductivity ($\Omega$), partial pressure of $CO_2$ ($pco$), carbonate content ($cac$), sulfate content ($soc$), chloride content ($clc$), acid acetic content ($acc$), temperature (*T*), pit initiation time ($t_{sd}$), and immersion time (*t*). A regression analysis was performed for each of the 512 possible combinations for the distribution of the nine variables that characterized the test. The best regression model among these 512 analyses was selected based on the highest value of the coefficient of determination ($R^2$) and is expressed as follows:

$$y_{max}(t) = Exp(\alpha_0 + \alpha_1 \Omega + \alpha_2 pco + \alpha_3 soc + \alpha_4 clc + \alpha_5 acc + \alpha_6 orp)(t - t_{sd})^{(\beta_0 + \beta_1 pH + \beta_2 cac + \beta_3 T)} \tag{18}$$

where $\alpha_i$ and $\beta_j$ are regression parameters that are shown in Table 5 in the study done by Velazquez and coworkers [46]. A sensitivity analysis was also performed to determine which variable exerted a greater influence on the internal corrosion defect growth caused by oilfield-produced water. The variables that exert more influence are pH, temperature, conductivity, redox potential, and carbonate ion content. For the sake of schematization,

the internal pitting corrosion growth under the average conditions of the synthetic oilfield-produced water is shown in Figure 6. Meanwhile, Figure 7 shows an image obtained by scanning electron microscopy (SEM), where the localized corrosion defect generated after immersion test of steel coupons in synthetic-produced water can have a diameter of several hundreds of micrometers. Figure 8 shows an image that exemplifies the damage caused by localized corrosion defects in a pipeline that transports oilfield-produced water.

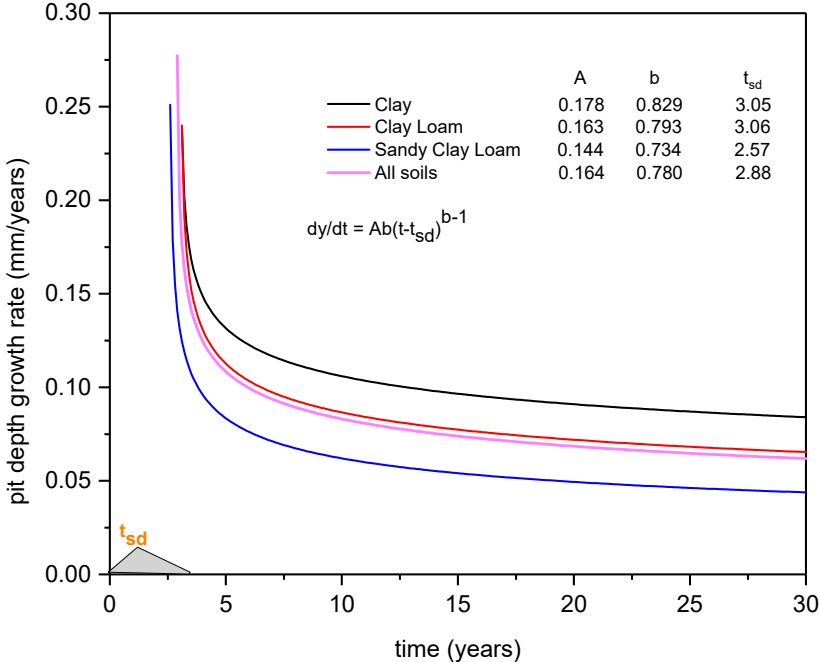

**Figure 5.** Evolution of pitting growth rate predicted for an average typical condition for soils in southern Mexico according to the study presented by Velázquez et al. adapted from [14].

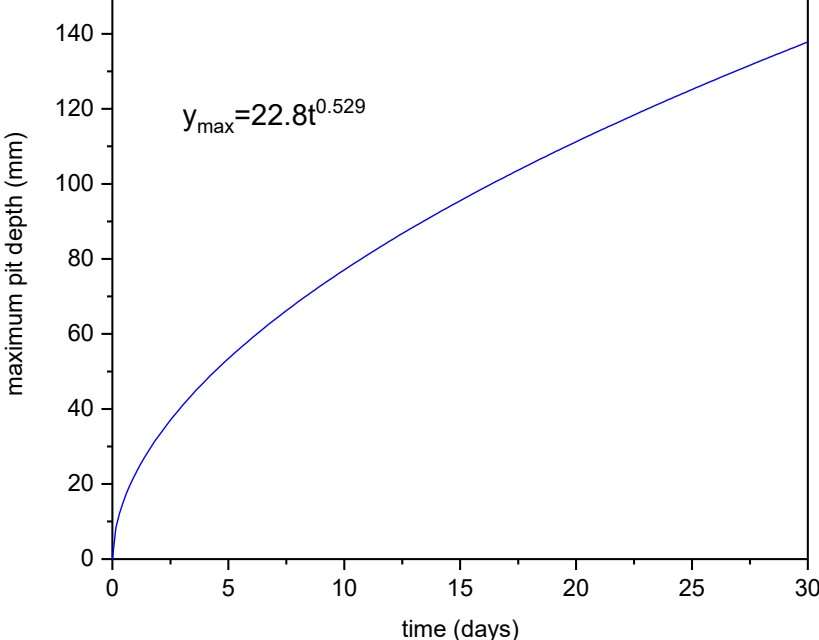

**Figure 6.** Maximum pit growth evolution as a function of immersion time of pipeline steel in synthetic-produced water according to the results obtained by Velázquez et al. reprinted with permission from ref. [46]. Copyright 2022 Elsevier.

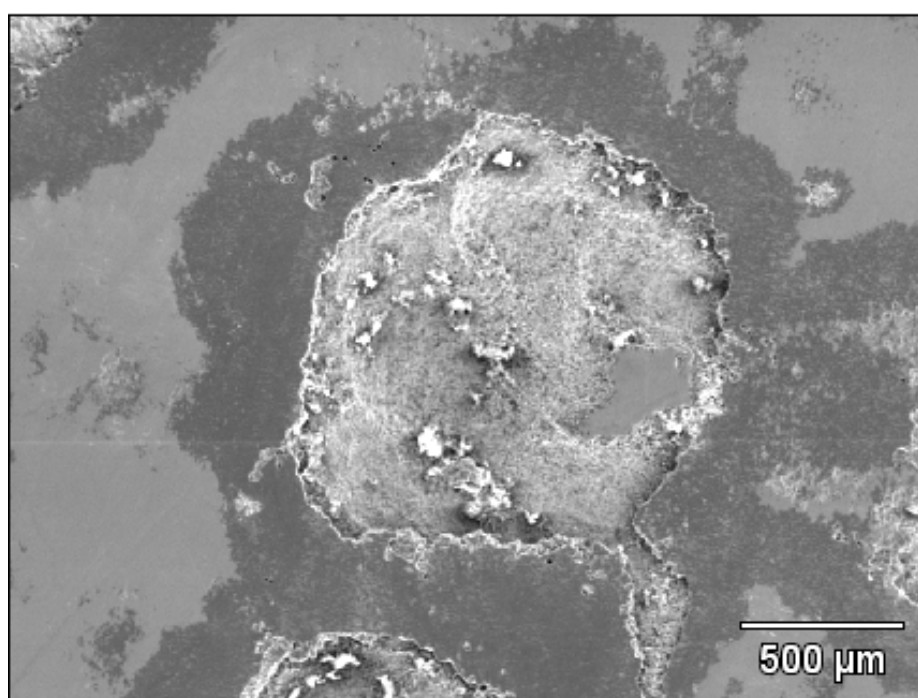

**Figure 7.** SEM image of localized corrosion defect after immersion test of API-5L-X60 steel under synthetic-produced water.

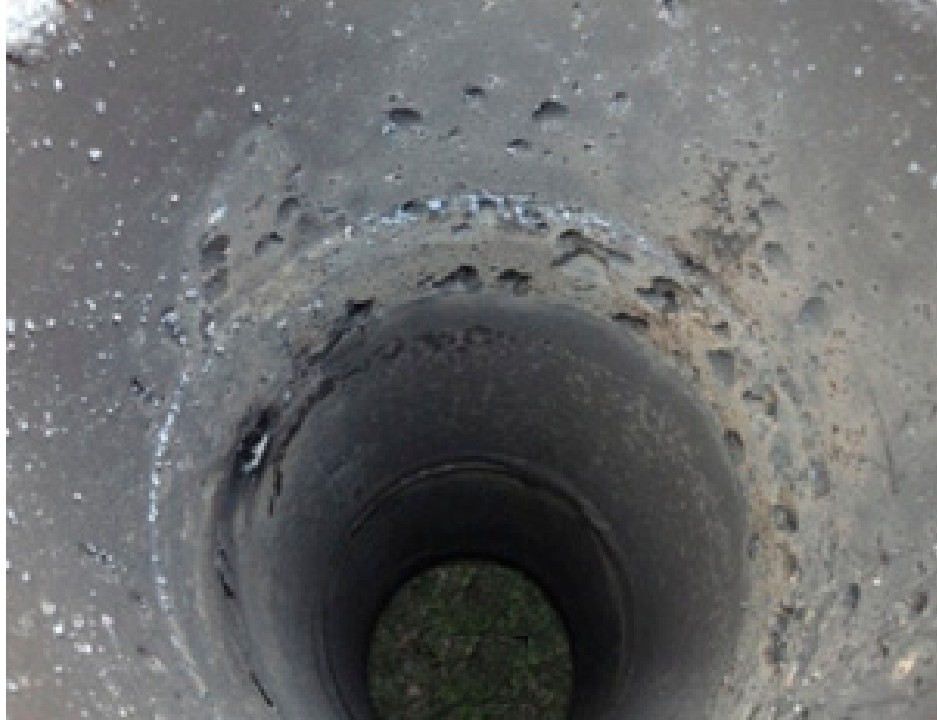

**Figure 8.** Internal localized corrosion defects in a pipeline that transports oilfield-produced water.

An important characteristic of these two models developed by Velazquez et al. is that they include the pit initiation time ($t_{sd}$), which is a value to be determined by regression analysis. The values of $t_{sd}$ indicated that the pit growth never starts immediately when the metal is in contact with the corrosive environment; it has an incubation period.

Using field data from wellheads, C. Ossai [62] proposed linear regression models to estimate the internal corrosion rate as a function of the operational parameters. The results for these models are shown in Equation (19a–d):

$$CR = \alpha_{11} + \beta_{11}T + \beta_{12}P_{CO_2} + \beta_{13}V_m + \beta_{14}pH \tag{19a}$$

$$CR = \alpha_{21} + \beta_{21}T + \beta_{22}P_{CO_2} + \beta_{23}V \tag{19b}$$

$$CR = \alpha_{31} + \beta_{31}T + \beta_{31}P_{CO_2} + \beta_{32}V_m \tag{19c}$$

$$CR = \alpha_{41} + \beta_{41}T + \beta_{42}P_{CO_2} \tag{19d}$$

where $CR$ is the corrosion rate, $T$ is temperature, $P_{CO_2}$ is the partial pressure of $CO_2$, $V_m$ is the mixed velocity of the fluid flowing through the wellhead, $V$ is the flow velocity of crude oil, and $pH$ is the pH value, whereas $\alpha_{ij}$ and $\beta_{ij}$ are the regression coefficients shown in Table 5 in Ossai's study [62]. This study led by Ossai, which was validated with field data, showed that approximately 26% of wellhead corrosion was provoked by the aforementioned operating parameters. Other variables such as organic acids, microbiological corrosion, erosion, condensation, and turbulence could be responsible for the rest of the corrosion damage.

Other applications of regression models in corroded oil and gas pipelines exist; however, the corrosion rate is only considered as an independent variable. For example, the regression model developed by K. Zakikhani and coworkers [63], where the corrosion rate and corrosion mechanism are input variables to estimate the time of pipeline failure, is quite useful for maintenance management. Al-Fakih et al. [64], performed an example in which the assessment of mild steel corrosion in 1 M hydrochloric acid used as an inhibitor was studied and the corrosion rate was considered an independent variable in the regression model.

The advantage of regression models is that it is simple for engineers to use this methodology to estimate the future corrosion defect depths in pipelines. It is not necessary to have deep mathematical knowledge to understand the application. On the other hand, the main disadvantage is that it cannot reproduce the random nature of the pitting corrosion phenomenon despite using a significant amount of data.

The possible future outlook from different viewpoints in regression analysis should include independent variables such as alloying element content in the steel, pipeline time commissioning, a more sophisticated method to determine the pit initiation time, and the slope of the soil on the section of pipe studied.

### 4. Stochastic and Random Walk Models

A couple of pioneering studies that used the stochastic nature of the localized corrosion defects in oil and gas pipelines were led by Howard Finley [65,66] using statistics of extremes in the 1960s. Nevertheless, Provan and Rodriguez III [67] proposed modeling the pitting corrosion growth as a Markov process two decades later. A stochastic process is called a Markov process if the present state of the process makes its future independent of the past. To model using Markov process theory, a metal plate can be divided into $N$ discrete Markov states. The corrosion defect depth at time ($t$) is represented by a discrete random variable $D(t)$ such that $P\{D(t) = i\} = p_i(t)$ with $i = 1, 2, \ldots, N$. Similarly, the transition probability can be defined as the probability of a transition from state i to j during the time interval $\tau$ to $t$:

$$P\{D_{\tau+t}(t) = j | D_\tau(t) = i\} = p_{ij}(\tau, t) \tag{20}$$

Provan and Rodriguez in 1989 modeled pitting growth without considering the pit generation process and proposed an expression for the intensities $\lambda_i(t)$ of the process with no physical meaning. Provan and Rodriguez did not discuss the method used to solve

the Kolmogorov differential equations (Equation (21)), which is necessary to reproduce their results.

$$\frac{dp_{ij}(t)}{dt} = \begin{cases} \lambda_{j-1}(t)p_{i,j-1}(t) - \lambda_j(t)p_{i,j}(t) \\ \\ \lambda_i(t)p_{i,i}(t) \end{cases} \tag{21}$$

where $\lambda_j = \lambda j(1 + \lambda t)/(1 + \lambda t^k)$.

Subsequently, using Equation (22), it is feasible to determine the probability of remaining in the state.

$$p_i(t) = \sum_{i=1}^{j} p_{ij}(\tau, t)p_i(\tau) \tag{22}$$

The model developed by Provan and Rodriguez [67] is based on the following assumptions for the transition scheme:

If the maximum pit depth on a certain area of observation is in state $j - 1$ at time $(t, t + \Delta t)$, it grows to state $j$ with likelihood $\lambda(j - 1)[(1 + \lambda t)/(1 + \lambda t^k)]\Delta t$, where $\lambda$ and $k$ depend on the characteristics of the corrosion environment.

The probability of increasing by more than one state in this interval is $0(\Delta t)$.

In Part II of the research conducted by Rodriguez and Provan [68], they highlighted the application of their stochastic model, using the Markov process, to estimate the reliability of deteriorating structures such as an oil pipeline system (the details of Part II of the research done by Rodriguez and Provan [68] are provided in the latter part of this study).

Another application of the Markov process theory in oil pipelines was presented by Hong [69], who used an analytical solution to the system of Kolmogorov's differential equations (Equation (21)) for the same homogeneous continuous type of Markov process and determine the process probability transition matrix to assess the probability of pipeline failure. Hong modeled the generation of new corrosion defects by a Poisson process and considered the uncertainty in defect detection by nondestructive inspection tools. Hong also explored the relationship between the maximum pit depth and load resistance ratio.

Another application of Markov process theory in oil and gas pipelines was proposed by S.A. Timashev and coworkers [70] in the first decade of the 21st century. They expressed a new model based on the use of a continuous-time, discrete-state pure birth homogeneous Markov process to stochastically describe the growth of localized corrosion defects. Timashev's research aimed to calculate the conditional probability of pipeline failure and to optimize the maintenance of operating pipelines. In their model, the intensities of the process were calculated by iteratively solving the proposed system of Kolmogorov's forward equations. Despite the significant contribution of Timashev et al. [70], some limitations were observed in their proposed model:

The relative complexity of the iterative method applied to estimate the transition intensities ($\lambda_i$) requires that the pipe wall thickness be divided into a few Markov states.

The use of time-independent transition rates ($\lambda_i$) suggests that their estimated values represent the average of the time-dependent intensities $\lambda_i(t)$ over the selected period. The time homogeneity condition implies that the elapsed time of corrosion damage in a given state is exponentially distributed. This also implies that the corrosion growth rate is treated implicitly as a constant, whereas the corrosion depth is treated as a linear function of the exposure time. Because of the nonlinearity of the localized corrosion defect process, these assumptions remain true only if the estimations are made for long exposure times over relatively short time spans.

Finally, the solution to the Kolmogorov equations applies only to a specific condition of a pipeline that undergoes repeated in-line inspections with the same technology.

In a more recent study, F. Caleyo and coworkers [71] also modeled the pitting corrosion growth in buried oil and gas pipelines considering a continuous-time and non-homogeneous linear growth (pure birth) Markov process. For a Markov process defined by the system of Equation (21), the conditional probability of transition from the $m$-th to the $n$-th state ($n \geq m$) in the interval $(t_0, t)$, that is, $P\{D(t) = n | D(t_0) = m\} = p_{mn}(t_0, t)$, can be obtained as follows [72]:

$$p_{mn}(t_0, t) = \binom{n-1}{m-1} e^{-\{\rho(t)-\rho(t_0)\}m} \left(1 - e^{-\{\rho(t)-\rho(t_0)\}}\right)^{n-m} \tag{23}$$

where

$$\rho(t) = \int_0^t \lambda(t') dt' \tag{24}$$

The identification of the transition probability function can be achieved by correlating the mean of the stochastic pit depth with the experimentally obtained deterministic mean. The stochastic mean can be expressed as follows [73]:

$$M(t) = n_i e^{\rho(t-t_i)} \tag{25}$$

In contrast, the deterministic mean of the maximum pit depth at time $t$ can be represented by the power-law model and can be written using Equation (14) [14]:

$$D(t) = A(t - t_{sd})^b \tag{26}$$

where $t_{sd}$ is the starting time of pitting corrosion.

Equating Equations (25) and (26), the function $\rho(t)$ can be computed as follows:

$$\rho(t) = ln(A(t - t_{sd})^b) \tag{27}$$

Therefore, the probability parameter $p_s = e^{-\{\rho(t)-\rho(t_0)\}}$ described in Equation (23) can be expressed as follows:

$$p_s = \left(\frac{t_0 - t_{sd}}{t - t_{sd}}\right)^b \tag{28}$$

If the transition probability function (Equation (23)) is known, then the distribution of the pit depth in the future can be estimated using Equation (22).

The observed probability distribution of the pit depth at $t_0$ was used as the initial corrosion damage distribution, $p_m(t_0)$. The transition probability function $p_{mn}(t_0, t)$, which is completely identified if the function $\rho(t)$ is known, would be accessible if a predictive model is available to relate the parameters $t_{sd}$ and $m$ in mathematical expression (28) to the physical and chemical characteristics of the soil [14,61]. This means that from the measurements of the in-line inspection and with the characterization of the local soil, it would be possible to determine the evolution of the pitting corrosion depth. Caleyo and colleagues illustrated the application of their Markov chain model using information from in-line inspections, as shown in Figure 9.

Timashev and Bushinskaya [74] described the growth of corrosion defects by Markov processes of pure birth and pure death type. These authors proposed a solution for the differential equation systems (21), which can be written as follows:

$$P_i(t) = \sum_{j=1}^i \mu_{ij} exp\{-\lambda_j t\} \; (1 = 1, \ldots, M), \tag{29}$$

where

$$\begin{cases} \mu_{11} = p_1^* \\ \\ \mu_{ij} = \begin{cases} \mu_{i-1,j} \frac{\lambda_{i-1}}{\lambda_i - \lambda_j}, i \neq j \\ p_i^* - \sum_{q=1}^{i-1} \mu_{iq}, i = j, \; i \leq k \\ -\sum_{q=1}^{i-1} \mu_{iq}, i = j, i \geq k \end{cases} \end{cases} \tag{30}$$

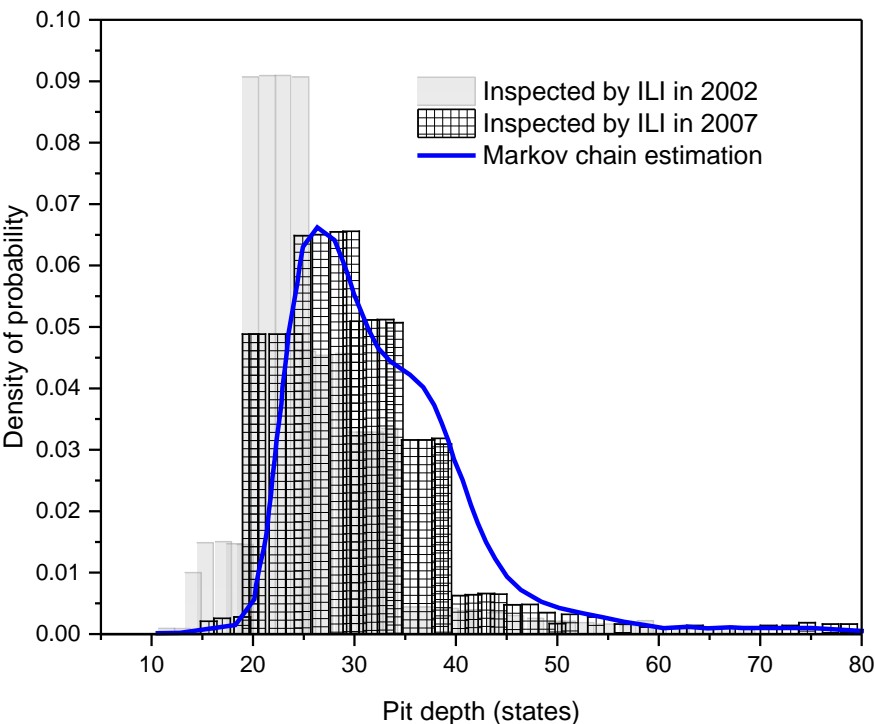

**Figure 9.** Results of the Markov chain modeling of pitting corrosion evolution from two recurrent inspections according to the research work led by Caleyo and Velázquez. Modified and turned into color using information presented in [71].

The unknown intensities $\lambda_i$ $(i = 2, 3, \ldots, M)$ are computed numerically by sequentially solving equations

$$P_i(t_1) = p_i = \sum_{j=1}^{i} \mu_{ij} exp\left[-\lambda_j(t_1)\right] \tag{31}$$

where $p_i = \frac{n_i(t_1)}{N(t_1)}$, $i = 2, \ldots, M$; $n_i(t_1)$ is the number of defects whose depths are in the $i$-th interval and $N(t_1)$ is the total number of defects detected in a pipeline at the initial time $t = t_1$.

Therefore, the model developed by S.A. Timashev and A.V. Bushinskaya [74] exemplified the application of this stochastic model to a pipeline to obtain satisfactory results. However, the solution of this model is purely based on mathematics and does not consider the variables involved in pipeline corrosion damage.

H. Wang et al. [75] proposed a novel methodology in which a hidden Markov field model and clustering methods were applied. This proposal is a statistical model in which it is assumed that the system to be modeled is a Markov process with unknown parameters. The purpose is to determine the unknown (or hidden, and hence the name) parameters of the string from the observable parameters. This methodology covers the conventional finite mixture model such that the spatial correlation of external corrosion sites can be considered. The methodology has been proven to classify corrosion defects using soil properties (resistivity; redox potential; and ionic content such as chloride, bicarbonate, carbonate, and sulfate) from field studies and location information from in-line inspections. The categorization performed revealed hidden patterns of corrosion damage in different segments along an oil pipeline. A stochastic simulation was used to test the proposed clustering approach. Wang's methodology was applied to an in-service pipeline to obtain acceptable results. The clustering analysis helps in showing the pattern of the soil properties and categorizing the corrosion defects by looking for homogeneous segments from a heterogeneous random field of soil properties. The incorporation of soil test and in-line inspection data is evidenced to be a better outline, which can be incorporated into a clustering-based inspection strategy that can improve maintenance policy.

Ossai et al. recently conducted a study that applied Markov process theory [76]. They developed a continuous-time non-homogeneous linear growth pure birth Markov model to corroded pipelines. The transition probability determined by Ossai et al. (2016) is described in Equation (32):

$$p_{n_0,n(t_0,t)} = \frac{(n-1)!}{(n_0-1)!(n-n_0)!} \left( \frac{t_0 - t_{sd}}{t - t_{sd}} \right)^{n_0} \left( \frac{t - t_0}{t - t_{sd}} \right)^{n-n_0} \tag{32}$$

where $p_{n_0,n(t_0,t)}$ is the transition probability of moving from state $n_0$ to $n$ in the time interval $(t_0, t)$. If the initial distribution of the corrosion defect depth is known using Equations (22) and (33), it is possible to estimate the future distribution of corrosion defect depths.

Ossai et al. generated a vector of random values of internal localized corrosion defect depths in pipelines, which were obtained using a linear multivariate regression model. This regression model considered some operational conditions of the fluid transmitted as independent variables, such as temperature, partial pressure of $CO_2$, flow rate, and pH. This linear regression model is expressed as follows:

$$y_{max}(t) = (\beta_0 + \sum_{i=1}^{m} \beta_i x_i)(t_i - t_{sd}) \tag{33}$$

where $y_{max}(t)$ is the maximum corrosion defect depth, $x_i$ are the operational parameters, $t_{sd}$ is the initiation time of the corrosion defect, and $\beta_i$ is the coefficient of the operational variables. The model expressed in Equation (33) considers that the corrosion rate has a linear behavior over time. A random number of 5000 runs were generated using the real operation conditions of the in-service pipelines. Using the information obtained from the simulated pit depth and the Poisson square wave process, Ossai et al. estimated the pipeline failure time and corrosion defect initiation time $t_{sd}$. After the simulation and estimation of the coefficients and initiation time, the transition probability function given in Equation (32) can be calculated. This Markov predicted model was tested with field data in both onshore and offshore oil and gas pipelines, and the results agreed. The transition probability function proposed by Ossai et al. is also a function of the defect initiation time, similar to the suggestion by Caleyo et al. [71] and described in Equation (23).

Other studies exist that used the Markov process theory to model pitting corrosion; however, these were performed using immersion tests in the laboratory. For example, Valor et al. [77] modeled the pitting corrosion, both initiation and growth, by stochastic theory using data from other authors [67,69,78,79]. The results obtained in this model were compared with the results of previous studies.

In another study published in 2010, Valor et al. applied the Markov chain model proposed by Caleyo et al. [71] to estimate the pitting corrosion depths generated in immersion tests on pipeline steel coupons. This means Markov theory is quite useful and applicable for both laboratory tests and in-service corroded oil and gas pipelines, as well as for research in materials engineering [80].

Other statistical techniques are used to model the growth of corrosion defects in pipelines as a random walk. Rafael Amaya-Gómez et al. [81] modeled the growth of corrosion defects based on a mixed Lévy process. This model combined the gamma and compound Poisson processes. They divided the corrosion degradation phenomena into three categories: (a) shock-based process, (b) progressive degradation, and (c) combined degradation. A shock-based degradation process occurs when discrete amounts of the system's capacity are detached at distinct points in time owing to impulsive and independent events (e.g., localized corrosion defects). These shocks are expected to randomly occur over time because of some mechanisms and two stochastic processes which define them: (a) interval time between shocks and (b) the damage of each shock. A special case of a shock mechanism is a compound Poisson process, and the interval time is distributed exponentially. The authors considered the shocks following a Poisson process with a rate $\lambda$. The progressive degradation process results from the ability to be uninterruptedly reduced at a rate that may change over time. The degradation model selected by Amaya-Gómez

et al. [81] was the gamma process. A gamma-process-based approach was proposed for use in degradation problems by van Noortwijk [82], and its application was further analyzed by S. Kuniewski et al. [83] and by J.C. Velázquez et al. [84]. The gamma process is stochastic with independent gamma-distributed growth, making it permanently random and monotonic. The gamma process with shape function $\nu(t) > 0$ and scale parameter $\zeta(t) > 0$ is a continuous-time stochastic process with the following characteristics:

$y(0) = 0$ with probability of 1;

$y(t) - y(r) \sim ga(\cdot | \nu(t) - \nu(r), \zeta)$;

$y(t)$ exhibits independent increments;

where $ga(\cdot | \nu, \zeta)$ is the gamma PDF with shape parameter $\nu$ and scale parameter $\zeta$, recalling that the random variable is the depth of the corrosion defect $y$. The gamma probability density function is expressed using Equation (34):

$$ga(y|\nu, \nu) = \frac{\zeta^\nu}{\Gamma(\nu)} y^{\alpha-1} exp\{-\zeta y\} \tag{34}$$

where $\Gamma(\nu)$ is the gamma function.

Finally, a combined degradation includes the gamma and compound Poisson processes. In addition, some of the main stochastic approaches used to model the degradation process associated with corrosion are presented. The mixed process used by Amaya-Gómez et al. is a Lévy process approach.

Amaya-Gómez et al. [81] conducted the first study that used the Gamma process to model pipeline degradation using a real case, including the maintenance cost, which is quite useful for the manager of a pipeline company.

In general terms, both stochastic models and random walks models have the advantage that can reproduce with high confidence the random nature of the localized corrosion defect growth. In addition, these kinds of models can incorporate the physical and chemical characteristics of the environment that is in contact with the pipeline. Therefore, it is feasible to estimate the maximum corrosion defect depths, which are the defects that threaten the pipeline's mechanical integrity. A limitation of these models is that specialized knowledge in stochastic processes and programming skills are necessary for their usage.

The current status and future outlook for stochastic and random walk models are covered in the present review. However, a key point in pitting corrosion modeling that is missing is to consider the stochastic nature of the pit starting time, which could bring great benefits.

## 5. Other Examples of the Use of Statistics in the Prediction of the Lives of Oil and Gas Pipelines

Other examples of estimating the life of a pipeline other than the aforementioned models exist. A recommended book detailing the mathematical principles involved in the estimations of the remaining life of pipes and vessels is titled *Introduction to Life Prediction of Industrial Plant Materials* [85]. This book explains how to handle the corrosion data for several deterioration mechanisms using extreme statistics starting from statistics theory. Moreover, to acquire knowledge on the recent applications of statistics in this topic, more industrial examples, or at least a new discussion of the well-known techniques, previous studies should be consulted. In this context, some studies are listed below to exemplify some uses of probability and statistics in the prediction of the life of oil and gas pipelines:

- "Probability Distribution of Pitting Corrosion Depth and Rate in Underground Pipelines: A Monte Carlo Study" by F. Caleyo et al. [15]. In this study, the probability distributions of the external-corrosion pit depth and pit growth rate were investigated in buried pipelines in a range of Mexican soils using Monte Carlo simulations. Information from previous studies [14,61] and the regression model already discussed were used to determine the best fit to the pitting depth and rate data for different future times. The distributions studied were Gumbel, Weibull, Fréchet, and generalized extreme values [86] (Gumbel, Weibull, and Fréchet distributions are special cases

from generalized extreme value distribution [86]; the mathematical expression that represents GEVD is shown in the latter part of this paper). It was observed that the means, variances, and shape parameters of these distributions differ significantly between soil types, and they were not completely constant for different exposure times. These differences developed more substantially as exposure time increased. However, after long exposure times, the distribution of the corrosion rate achieved a relatively constant yet slightly decreasing mean and variance. The probabilistic corrosion rate distributions provided in this study can be used to accurately estimate the reliability evolution of oil and gas pipelines rather than reclining the conservative average pit growth rates in existing studies. The predicted probability function that describes the distribution of the pit depth, computed by Caleyo et al. [15], is as follows:

$$f(y)_{ft} = \int_0^\infty g(v) f_{ii}(x - v\delta)_{last} dv \tag{35}$$

where $f(y)_{ft}$ is the estimated PDF of the depth of the corrosion pit at a later time $t + \delta$, $g(v)$ is the PDF of the corrosion rate, $\delta$ is the time elapsed between two inspections, and $f_{ii}(x)_{last}$ is the function determined in the final inspection.

- "Stochastic Modelling of Corrosion Damage Propagation in Active Sites from Field Inspection Data" by Alamilla and Sosa [16]. In this study, the PDF of the depths of corrosion damage of pipeline systems was computed, and four models to calculate the velocities of corrosion damage at localized defects were proposed. Each of these models is described as follows:

  I.  A model that considers the generation of corrosion defects following a Poisson process was proposed. The corrosion rate is expressed as follows:

$$f(v)_i = \frac{dm_i}{v^2} f_{X_i}(t_1 - dm_i v^{-1}) \tag{36}$$

  where $f(v)$ is the PDF for the corrosion rate, $dm$ is the measurement of the depth of the corrosion defect, $t$ is the time of inspection, and $f_X$ is expressed as follows:

$$f_{X_i}(x) = \frac{n}{t_1} \binom{n-1}{i-1} \left(\frac{x}{t_1}\right)^{n-i} \left(1 - \frac{x}{t_1}\right)^{i-1} \tag{37}$$

  where $n$ is the number of measurements of depths of corrosion defects.

  II. In the second model, it was considered that two inspections were performed in oil and gas pipelines at different time instants and the defects were identifiable in both inspection reports. In addition, the generation of corrosion defects was not considered. The corrosion rate is estimated to have less variability. In this case, the corrosion rate is represented as follows:

$$f_V(v) = \frac{1}{n} \sum_{i=1}^n \left[\phi_{V_i}(0)\right]^{1-I_{V_i}} \left[\Delta_t \varphi_{V_i}(v\Delta t)\right]^{I_{V_i}} \tag{38}$$

  where $\varphi_{V_i}$ is the normal PDF, with mean $\mu_i = d_{1_i} - d_{0_i}$ and standard deviation $\sigma_i = \left(\sigma_{\varepsilon_1}^2 - \sigma_{\varepsilon_2}^2\right)^{1/2}$; $\sigma_{\varepsilon_i}$ is the standard deviation of the measurements; $\phi_{V_i}(0)$ is the normal cumulative function of $\varphi_{V_i}(\cdot)$ at $v = 0$; $\Delta_t = t_1 - t_0$; $I_{V_i}$ is the indicator of the $i$-th corrosion defect that is equal to 1 if $0 < d < w_0$ and equal to 0 if $d = 0$; and $w_0$ is considered as the initial thickness of the pipe wall.

  III. Because the main disadvantage of the second model is the identification of corrosion defects in two consecutive inspections, a third model considers the PDF of the depths of corrosion defect of a pipeline system $f_D(d(t_1))$ and $f_D(d(t_0))$, related to inspections in $t_1$ and $t_0$, respectively. This third model, proposed by Alamilla and Sosa, is expressed as follows:

$$f_V(v) = \frac{1}{n_0 n_1} \sum_{i=1}^{n_1} \sum_{j=1}^{n_2} \left[\phi_{V_{ij}}(0)\right]^{1-I_{V_i}} \left[\Delta_t \varphi_{V_i}(v\Delta t)\right]^{I_{V_i}} \tag{39}$$

where $n_0$ and $n_1$ are the number of corrosion defects reported in each inspection.

IV. The fourth model is based on Bayes' theorem, and it is used to update the propagation function (corrosion rate function) by including new measurements from sequential inspections. The updated propagation function is expressed as follows:

$$f_V''(v) \propto f_V(v) \left( \left[\phi_{V_i}(0)\right]^{1-I_V} \left[\Delta_t \int_0^{w_0} f_{D_1}(v\Delta_t + x) f_{D_0}(x) dx \right]^{I_V} \right) \tag{40}$$

where $f_V''$ is the updated probability function of the corrosion rate.

- "Modelling Steel Corrosion Damage in Soil Environment" by Alamilla et al. [87]. A model to estimate the propagation of the localized corrosion damage in buried oil and gas pipelines was developed considering the physical and chemical soil characteristics. This model offers a satisfactory description of the evolution of corrosion damage and minimizes most of the inconveniences of the power law used (Equation (7)). The variability of the depths of corrosion defects was satisfactorily represented by the Gumbel PDF. The depth of the corrosion defect can be estimated as follows:

$$y(t) = v_p t + \frac{v_0 - v_p}{q_0}[1 - \exp(-q_0 t)] \tag{41}$$

where $v_0 = v(0)$, $q_0$ is a constant to be determined, and $v_p$ can be determined as follows:

$$v_p = C_0 exp[-(q_1 pH + q_2 re + q_3 rp + q_4 pp] \tag{42}$$

where $C_0$ is the scale factor and $q_i$ are constants related to the soil characteristics. To calibrate this model, Alamilla et al. [87] used a Mexican database, a National Bureau of Standards database, and the New York database.

- "Stochastic Process Corrosion Growth Models for Pipeline Reliability" by Felipe Alexander Vargas Bazán and André Teófilo Beck [88]. A nonlinear model was proposed, in which the corrosion rate was studied as a Poisson square wave process. Instead of proposing a parameterized stochastic process by considering the parameters of the power-law equation as random variables, the proportionality factor of the power-law function (Equation (7)) is exhibited as a Poisson square wave process. This tolerates temporal uncertainty in the growth of corrosion defect to be characterized but continues to grow. The authors defined four models for the growth of corrosion defects. The models are listed in Table 1.

**Table 1.** Characteristics of the four models used by Bazán and Beck [88].

| Model | Characteristic | Distributions Used | Description |
|---|---|---|---|
| Linear random variable | • Growth rate | • Gamma | It is a simple representation of the corrosion defect rate ($v$) as a random variable. It can be estimated as follows:<br><br>$$y(t) = y_0 + v\Delta t \qquad (43)$$<br><br>where $y(t)$ is the defect depth to be estimated, $y_0$ is a previous defect depth, $v$ is the corrosion rate, and $\Delta t$ is the interval |
| Linear stochastic process | • Growth rate pulse heights<br>• Growth rate pulse durations | • Gamma<br>• Exponential | The authors proposed modeling the growth rate of the corrosion defect as a Poisson square wave process with stationary and independent increments (pulse heights). The pulse height ($Y_i$) and time durations ($t_{bi} = t_{i+1} - t_i$) are both characterized as random variables. The pulse durations are described as an exponential random variable with parameter $\lambda$. Conversely, pulse heights are represented using a gamma distribution. The following equation describes this process:<br><br>$$y(t_{i+1}) = y(t_i) + Y_i(t_{i+1} - t_i) \qquad (44)$$<br><br>for $i = 1, 2, \ldots, n$. |
| Nonlinear random variable | • Proportionality factor<br>• Exponent factor | • Gamma<br>• Lognormal | In this case, the authors used the power-law model proposed by Velázquez et al. [14] and Caleyo et al. [15] (Equation (14)). Bazán and Beck modeled considering both the proportional and exponential factors as random variables, considering the initiation time of 2.88 years. The proportionality factor is supposed to follow a gamma distribution. The exponent is supposed to follow a lognormal distribution. |

**Table 1.** *Cont.*

| Model | Characteristic | Distributions Used | Description |
|---|---|---|---|
| Nonlinear stochastic process | <ul><li>Proportionality factor pulse heights</li><li>Proportionality factor pulse durations</li><li>Exponential factor</li></ul> | <ul><li>Gamma</li><li>Exponential</li><li>Lognormal</li></ul> | This model emerges as a combination of the nonlinear random variable model and the linear stochastic process model. The proportional factor of the power law (see Equation (14)) is represented by a Poisson square wave process, with pulse height $Y_i$ and durations $t_{bi}$. Both exponential and gamma distributions are used to represent the arrival and new pulses and pulse intensities, respectively. Similarly, the lognormal distribution is used to represent the exponential factor of the power law. The increment in the size of corrosion defect is given as follows: $$y(t_{i+1}) = y(t_i) + Y_i\left[(t_{i+1} - t_0)^b - (t_i - t_0)^b\right] \tag{45}$$ for $i = 1, 2, \ldots, n$. |

The nonlinear stochastic process model provided the best result after applying these four models to an in-service pipeline.

- "The Negative Binomial Distribution as a Model for External Corrosion Defect Counts in Buried Pipelines" by Valor et al. [89]. In this study, the statistical analysis of real corrosion information from 50 buried oil and gas pipelines operating in Mexico led to the conclusion that the negative binomial (NB) distribution provides a correct description of corrosion defect counts, and the authors discussed the origin of this distribution for this phenomenon. Therefore, the corrosion defect count or corrosion defect density is a random variable that is independent and identically distributed, considering the number of corrosion defects per unit area. The causes determining NB as the distribution for defect counts are associated with three processes: gamma–Poisson mixture, compound Poisson process, and Roger's process. Unlike other studies, where the number of corrosion defects is modeled as a Poisson process, the defects are randomly distributed and do not interact with each other, and the NB distribution allows representation of cluster corrosion defects that are more realistic in buried pipelines because of the heterogeneous conditions of the soil. Here, the NB distribution $NB(\mu, \kappa)$ is given as follows:

$$P_{NB}(x) = \frac{\Gamma(\kappa + x)}{x!\Gamma(k)}\left(1 + \frac{\mu}{\kappa}\right)^x \left(\frac{\mu}{\mu + \kappa}\right)^n \tag{46}$$

where $\Gamma(\cdot)$ is the gamma function and $\mu$ and $\kappa$ are the location and shape parameters, respectively. Another parameterization of NB takes the following form:

$$P_{NB}(x) = \frac{\Gamma(n + x)}{x!\Gamma(n)}(1 - p)^x p^n \tag{47}$$

where $n = \kappa$ and $p = \kappa/(\kappa + \mu)$.

After the statistical analysis of 50 buried oil and gas pipelines, it was shown that the Poisson distribution should be rejected as a correct model to represent the external corrosion defect count because the defects appear in clustered patterns. Valor et al. [89] also demonstrated that the gamma–Poisson process leads to an NB distribution that offers a good fit to the vertical defect count data. It was confirmed that the compound Poisson process is also a source of NB distribution. This occurs because the number of external defects in clusters of fixed lengths that are randomly distributed on the pipeline follows a logarithmic series distribution, whereas the spaces between clusters are represented by an exponential distribution. According to Valor et al. [89], it is feasible to postulate that Roger's clustered process is responsible for the observed NB distribution of external corrosion defect counts. Roger's process could explain that the birth of a new pit increases the likelihood that new pits originate in the surrounding area, leading to the creation of new clusters.

Once the characteristics of localized corrosion observed data have been fitted to theoretical probability density functions, it is feasible to estimate future corrosion defect depths (Caleyo et al. [15,16]), predict the actual corrosion defect counts (Valor et al. [89]), or compare the different corrosion rate models (Bazán and Beck [88]). With this theoretical function already fitted, one can simplify the analysis besides estimating the future characteristics of the corrosion defects in a pipeline.

## 6. Pipeline Reliability Estimations

Practically, all pipeline reliability estimations for corroded pipelines consider as a major assumption the randomness of load and resistance parameters established by the limit state function (LSF) [22,90]:

$$z = p_f - p_{op} \tag{48}$$

where z is positive for safe corrosion defects ($p_f > p_{op}$) and negative for unsafe defects, $p_f$ is the pipeline failure pressure, and $p_{op}$ is the working pressure. The LSF depends on the same variables as $p_f$. Thus, a complete form of the LSF can be defined by the following expression [22]:

$$z_1\left(p_f, t\right) = p_f(DD, th, YS \text{ or } UTS, y(t), l(t)) - p_{op} \tag{49}$$

where $D$ is the pipeline diameter, $th$ is the pipe wall thickness, $YS$ is the material yield strength, $UTS$ is the ultimate tensile strength, $y$ is the depth of the corrosion effect, and $l$ is the length of the corrosion defect. Because of the corrosion defects, characteristics evolve over time; therefore, these two variables are, in turn, a function of time.

A second limit state function can be defined as the difference between 80% of the thickness of the pipe wall (maximum allowable value) and the depth of corrosion defect. This is represented as follows [22]:

$$z_2(y, t) = 0.8th - y(t) \tag{50}$$

For localized corrosion defects that are presented in a pipeline, Equation (49) gives the limit state function for a local burst failure; meanwhile, Equation (50) indicates the limit state function for perforation. The composite failure probability (total probability of failure) can be estimated using Equation (51) [22,91]:

$$z_T(t) = z_1(p_f, t) + z_2(y, t) - z_1(p_f, t)z_2(y, t) \tag{51}$$

For an oil and gas pipeline with $n$ independent corrosion defects, the probability of pipeline failure can be computed as follows [22]:

$$PF_{pipe} = 1 - (1 - z_{T1}(t))(1 - z_{T2}(t)) \ \dots \ (1 - z_{Tn}(t)) \tag{52}$$

One methodology to estimate the probability of pipeline failure is the Monte Carlo method. Therefore, using a simulation method (i.e., the inverse CDF method [92], which consists in finding a random number from the inverse of a cumulative distribution function [92]: $\mathbf{Y} = \mathbf{F}_{\boldsymbol{\pi}}^{-1}(\mathbf{u})$), it is feasible to find a random number for each independent variable involved in Equations (50) and (51) and compute these two limit state functions. Other pioneering work in estimating pipeline reliability using the Monte Carlo method was performed by Caleyo et al. [22], where the results obtained by this method were compared with the outcomes found by the first-order second-moment (FOSM) method and first-order method (FOM). These three algorithms (Monte Carlo, FOSM, and FOM) produced similar results. Caleyo et al. [22] assessed the effect of different pressure models (e.g., B31G, modified B31G, Shell-92, DNV-92, and Batelle), and observed that for long periods, the Shell-92 and B31G models presented the highest and the lowest failure probabilities, respectively.

The Monte Carlo algorithm was also used by Valor et al. in 2013 to estimate the reliability evolution in an underground pipeline. Reliability evolution is needed because corrosion defects grow and consequently change their dimensions. Valor et al. [93] compared different distributions of corrosion rates derived from various corrosion growth models and their influence on reliability estimations. The distributions used are based on the NACE-recommended corrosion rate, corrosion rate derived from the linear corrosion rate model, time-dependent and time-independent corrosion rates, and corrosion rate derived from the Markov chain model discussed in the present review and detailed in [71]. The details of the distribution of each corrosion rate used are described as follows to estimate the depth of corrosion defects from previous in-line inspection information:

- Single-value corrosion rate. Valor et al. used the NACE-recommended value for the corrosion rate in pipelines, which is 0.4 mm/year [94]. This value of corrosion rate was used for each defect found in the in-line inspection to obtain future pit depths.

- Linear rate model. It is assumed that each depth of corrosion defect evolves at the same rate at any moment. The corrosion rate of the corrosion defects can be calculated using the following equation:

$$\dot{y}(t) = \frac{y(t_1)}{t_1} \tag{53}$$

  where $y(t_1)$ is the depth of corrosion defect measured at the first inspection and $t_1$ is the full life until the first inspection. To use this value, the initial defect depth $y(t_{ini})$ is influenced by the corrosion rate expressed by Equation (53) and the time elapsed between the two inspections. The obtained defect depth was represented by a histogram to yield a PDF.
- Time-dependent generalized extreme value distribution (GEVD) model. This model uses a time-varying corrosion rate distribution proposed by Caleyo et al. [15], which was also discussed in the present study. Caleyo et al. showed that the localized corrosion growth rate in underground pipelines could be represented by a GEVD [86]:

$$G(v) = exp\left\{ -\left[ 1 + \zeta\left( \frac{v - \dot{\mu}}{\dot{\sigma}} \right) \right] \right\} \tag{54}$$

  where $\zeta$, $\dot{\mu}$, and $\dot{\sigma}$ are the shape, scale, and location parameters, respectively. The values of these parameters change over time.
- Time-independent GEVD model. This is similar to the preceding model, excluding the fact that the GEVD parameters of the GEV distribution are assumed as constants and equal to the parameters at the time of the previous inspection.

For the time-dependent and time-independent GEVD models, the convolution mathematical expression represented by Equation (35) was used to estimate the depths of future defects.

Markov model. This model was also described in a previous section of this paper and was developed by Caleyo et al. [71]. To use this Markov model, only the distribution of the initial depth of the pit and soil characteristics must be known. The soil characteristics are necessary because the parameter $b$ from Equation (26) needs to be determined (parameter b is the exponent shown in Equation (14) and is a function of some soil characteristics according to J.C. Velázquez et al. [14]). These soil characteristics are the pipe-to-soil potential, soil water content, soil bulk density, and pipeline coating type. From Equations (14) and (23), it was feasible to obtain another equation to determine the distribution of the corrosion rate:

$$f(b; t_0, t) = \sum_{m=1}^{N} p_m(t_0) \begin{pmatrix} m + b(t - t_0) - 1 \\ b(t - t_0) \end{pmatrix} \left( \frac{t_0 - t_{sd}}{t - t_{sd}} \right)^{bm} \left[ 1 - \left( \frac{t_0 - t_{sd}}{t - t_{sd}} \right)^{b} \right]^{b(t - t_0)} (t - t_0) \tag{55}$$

where $N$ is the total of Markov states and $t_0$ corresponds to the time at which $p_m(t_0)$ is observed or the time of the initial inspection.

Valor et al. [93] used information provided for an in-line inspection performed in 1996 in a service pipeline. The defect depths found in this inspection were influenced by each aforementioned model, and the results were compared with a second inspection performed 10 years later. The results obtained after applying these five models are shown in Figure 10.

Because reliability estimations evolve, Valor et al. [93] estimated the reliability of the pipeline using a PCORR model and the same methodology proposed by Caleyo et al. [22] using the Monte Carlo algorithm.

The annual probability of failure is estimated as follows:

$$PoF_i^{ann}(t_j, t_{j+1}) = \frac{PoF_i^{ann}(t_{j+1}) - PoF_i^{ann}(t_j)}{1 - PoF_i^{ann}(t_j)} \tag{56}$$

where $t_j$ represents the specific year under analysis.

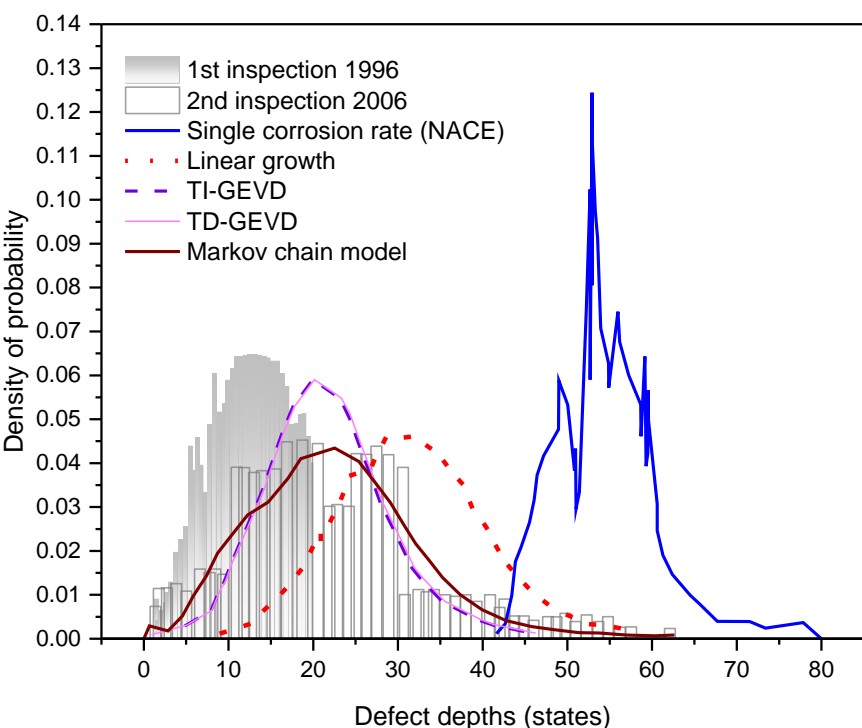

**Figure 10.** Exemplification of corrosion defect depth distributions obtained by five theoretical models and their comparison with empirical data obtained by ILI. This figure was modified (histogram bin width and colors in lines and bins were changed) using data from research conducted by Valor and coworkers reprinted with permission from ref. [93]. Copyright 2022 Elsevier.

If the corrosion defects are considered to be independent, the upper bound of the annual probability of failure for the entire pipeline can be estimated using the following expression:

$$PoF(t_j, t_{j+1}) = 1 - \prod_i \left[1 - PoF_i^{ann}(t_j, t_{j+1})\right] \tag{57}$$

In addition, the pipeline's annual failure index $\dot{\lambda}$ at the end of the *j*-th year can be computed as follows:

$$\dot{\lambda}(t_j, t_{j+1}) = \frac{\sum_{i=1}^{N_{def}} PoF_i^{ann}(t_j, t_{j+1})}{l_{pipe}} \tag{58}$$

where $l_{pipe}$ is the length of the pipeline (km). Hence, $\dot{\lambda}$ has units of failure per kilometer per year. Valor et al. [93] estimated the evolution of the failure index for the studied pipeline. They found that the time-independent GEVD model generated the most conservative annual failure index. Conversely, the single-value corrosion rate estimated a less conservative annual failure index. The annual failure index obtained for all models studied by Valor et al. is shown in Figure 11.

Valor et al. [93] concluded that the Markov model proved to be the best choice among the five models selected to estimate the evolution of pipeline reliability.

In addition, Caleyo et al. conducted a study that used the Monte Carlo algorithm, titled "On the Estimation of the Probability of Failure of Single Corrosion Defects in Oil and Gas Pipelines" [95], where they used the data on real-scale burst tests published by Pipeline Research Council International to estimate the probability of failure of pipe sections. All these pipe sections failed during the test, meaning the probability of failure associated with each defect should be sufficiently large or close to 1. Nevertheless, there are several cases in which the probability of failure is too small to be considered a risk. Therefore, the authors recommend interpreting the estimation of the probability of failure with extreme caution

because they observed the probability of failure values to be exceptionally low, especially in failure models that employed UTS as an independent variable.

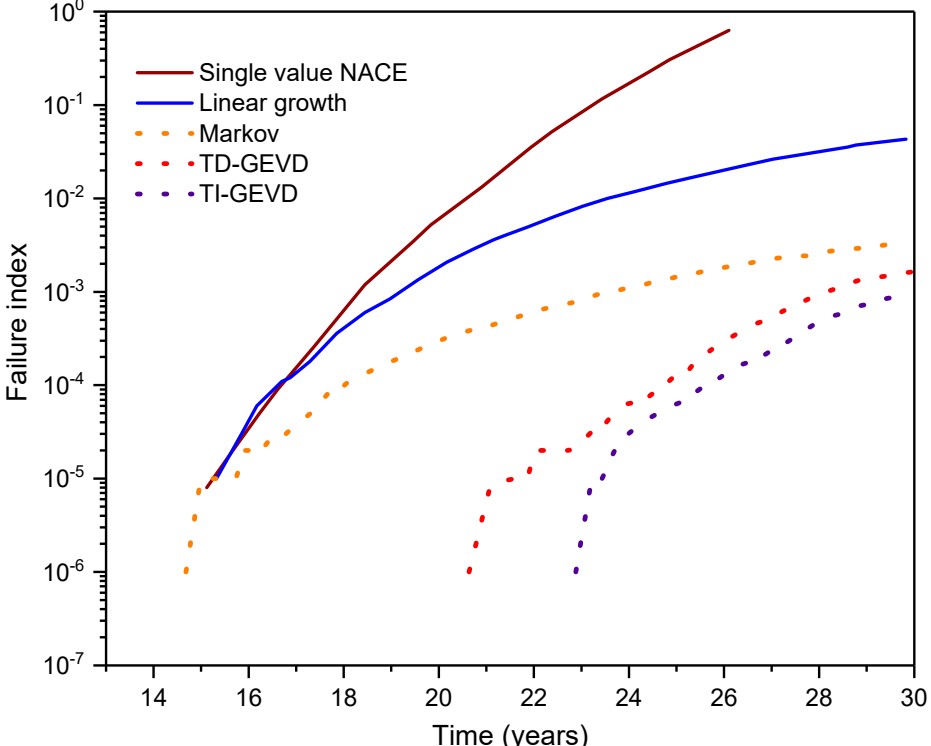

**Figure 11.** Evolution of the failure index in the pipeline studied and inspected by ILI and the prediction using five corrosion models. This figure was modified (color in lines was changed) using data from research conducted by Valor and coworkers reprinted with permission from ref. [93] Copyright 2022 Elsevier.

Recently, González-Arevalo et al. [23] aged pipeline steel using an artificial method to determine the changes in the mechanical properties of pipeline steel. Subsequently, they estimated the failure pressure of the pipeline for pipe sections by considering the changes in the mechanical properties owing to material aging. They observed that the changes in the probability of pipeline failure estimation could be considerable because of the changes in steel aging. Figure 12 illustrates these changes for a pipe section with a single defect for different models for estimating the probability of pipeline failure. The models for estimating pipeline failure pressure used by González-Arevalo et al. [23] are presented in Table 2. Other studies discussed in this paper also include some of these models to be performed [22,93,95].

Monte Carlo simulation outperforms other methods in estimating the pipeline probability of failure in that it is a technique of greater mathematical simplicity, it takes into account the randomness of each independent variable involved, and it is relatively easy to update the calculation algorithm if necessary. Likewise, it can be complemented with other methods such as the convolution of random variables, obtaining satisfactory results from a reliability engineering point of view.

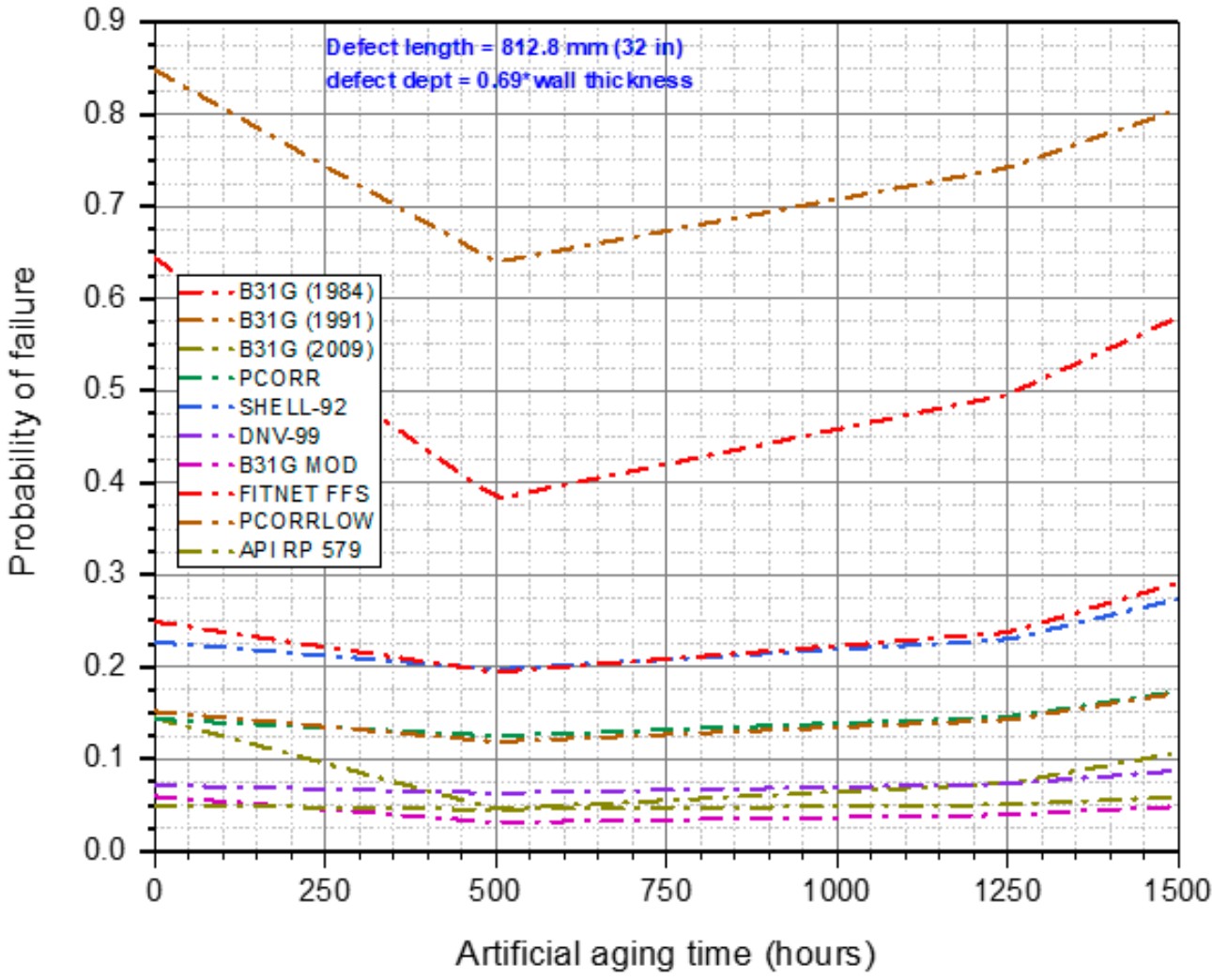

**Figure 12.** Pipeline failure pressure evolution for different artificial aging times for pipeline steel. This figure was modified using the data generated by the research led by González-Arevalo and Velázquez with permission from ref. [23] Copyright 2022 Elsevier.

**Table 2.** Pipeline failure pressure models.

| Model and References | Authors | Expression | Bulging Factor |
|---|---|---|---|
| ASME B31G-1984 [96] | ASME | If $\frac{L^2}{DDth} \leq 20$, $$P_b = 1.11\frac{2th\sigma_y}{DD}\left(\frac{1-\frac{2y}{3th}}{1-\frac{2y}{3tM}}\right)$$ If $\frac{L^2}{DDth} > 20$, $$P_b = 1.11\frac{2th\sigma_y}{DD}\left(\frac{1-\frac{y}{th}}{1-\frac{y}{thM}}\right)$$ | $M = \sqrt{1 + 0.6275\frac{L^2}{DDt} - 0.003375\left(\frac{L^2}{DDth}\right)^2}$ |
| ASME B31G-1991 [97] | ASME | If $\frac{L^2}{DDth} \leq 20$, $$P_b = 1.11\frac{2th\sigma_y}{DD}\left(\frac{1-\frac{2y}{3th}}{1-\frac{2y}{3thM}}\right)$$ If $\frac{L^2}{DDth} > 20$, $$P_b = 1.11\frac{2t\sigma_y}{DD}\left(1 - \frac{y}{th}\right)$$ | $M = \sqrt{1 + \left(0.8\frac{L^2}{DDth}\right)}$ |
| ASME B31G-2009 [98,99] | ASME | $$P_b = 1.11\frac{2\sigma_y th}{DD}\left(\frac{1-0.85\frac{y}{th}}{1-0.85\frac{y}{thM}}\right)$$ | If $\frac{L^2}{Dt} \leq 50$, $M = \sqrt{1 + 0.6275\frac{L^2}{DDth} - 0.003375\frac{L^4}{DD^2th^2}}$ If $\frac{L^2}{Dt} > 50$ $M = 3.3 + 0.032\frac{L^2}{DDth}$ |
| *Modified* ASME B31G [100] | ASME J.F. Kiefner P.H. Vieth | $$P_b = \frac{2(\sigma_y+68.95\text{ MPa})th}{DD}\left(\frac{1-0.85\frac{y}{th}}{1-0.85\frac{y}{thM}}\right)$$ | If $\frac{L^2}{Dth} \leq 50$, $M = \sqrt{1 + 0.6275\frac{L^2}{DDth} - 0.003375\frac{L^4}{DD^2th^2}}$ If $\frac{L^2}{Dth} > 50$ $M = 3.3 + 0.032\frac{L^2}{Dth}$ |
| Shell-92 [101] | Shell F.J. Klever G. Steward C.A.C. van der Valk | $$P_b = \frac{1.8t\sigma_u}{DD}\left(\frac{1-\frac{y}{th}}{1-\frac{y}{tM}}\right)$$ | $M = \sqrt{1 + 0.805\frac{L^2}{DDth}}$ |

**Table 2.** *Cont.*

| Model and References | Authors | Expression | Bulging Factor |
|---|---|---|---|
| PCORR [102,103] | Batelle<br>B.N. Leis<br>D.R. Stephens | $P_b = \frac{2t\sigma_u}{DD}\left(1 - \frac{y}{th}M\right)$ | $M = 1 - exp\left(-0.157\frac{L}{\sqrt{DD(th-y)/2}}\right)$ |
| DNV RP F101 [104] | Det Norske Veritas (Norway)<br>BG Technology (Canada) | $P_b = \frac{2t\sigma_u}{DD-t}\left(\frac{1-\frac{y}{th}}{1-\frac{y}{tM}}\right)$ | $M = \sqrt{1 + 0.31\frac{L^2}{DDth}}$ |
| API RP 579 [105] | API | $P_b = \frac{2t\sigma_u}{0.9DD}\left(\frac{1-\frac{y}{th}}{1-\frac{y}{tM}}\right)$ | $M = \sqrt{1 + 0.31\frac{L^2}{DDth}}$ |
| FITNET FFS [106] | European Fitness for Service<br>Network<br>E. Seib et al. | $P_b = \frac{2th\sigma_u(1/2)^{65/\sigma_y}}{DD-t}\left(\frac{1-\frac{y}{th}}{1-\frac{y}{tM}}\right)$ | $M = \sqrt{1 + 0.8\frac{L^2}{DDth}}$ |

For the expressions shown in Table 2, the variables involved are burst pressure ($P_b$), pipe wall thickness ($th$), pipe diameter ($DD$), corrosion defect depth ($y$), corrosion defect length ($L$), steel UTS ($\sigma_u$), and steel yield strength ($\sigma_y$).

## 7. Bayesian Data Analysis in Corroded Oil and Gas Pipelines

In oil and gas pipelines, all information on the inspection is not available, especially if the pipelines are non-piggables [107] (a non-piggable pipeline is a pipeline or section of pipeline in which it is not possible to send a pig device to perform an in-line inspection; a pipeline may be non-piggable because of extreme bends or changes in diameter [107]). In such pipelines, statistical techniques are of paramount importance because they allow estimation of the reliability and the remaining life with a certain confidence level. Bayesian data analysis (BDA) allows the computation of the probability of any particular value for a model parameter, addressing issues that could only be studied indirectly using the traditional statistical approach through the practice of random variable statistics. BDA handles situations where incomplete information is available, providing a liable and desired extension of the real information for extensive use. Therefore, this section explains the application of BDA to non-piggable pipeline systems in Mexico. The results of this study are presented in two papers [17,18], and the results are summarized to illustrate its application in oil and gas pipelines. First, Valor et al. led an extensive field survey in gathering pipeline systems [18]. This first part of the work helped in determining true values for the density of corrosion defects and for the sizing of the corrosion defects (length and depth). Similarly, the development of a statistical analysis of the characteristics of the studied pipeline was feasible (pipe diameter, pipeline length, and thickness of the pipe wall). Information on the depth and length of the corrosion defect was fitted to a GEVD (Equation (54)). The histograms of the measured depth and length of corrosion defect are shown in Figure 3a,b in [18]. The results of the use of the Kolmogorov–Smirnov (K-S) test confirmed the correctness of the selection of this distribution. A similar process was performed to determine the corrosion defect density. The number of defects per unit length of the pipeline was estimated by statistically studying the number of defects found in each ditch dug in the analyzed pipelines. The results show that the corrosion defect density per ditch is distributed as an NB distribution (Equation (47)). The parameters for the GEV distribution for the length and depth of corrosion defect, as well as the parameter for NB distribution, are shown in Table 3.

**Table 3.** Parameters of the fitted distributions for corrosion defect depth, length, and density of defects in the zone studied in Mexico [18].

| Characteristic | Units | Distribution | Parameters | | |
|---|---|---|---|---|---|
| | | | $\dot\mu$ | $\dot\sigma$ | $\zeta$ |
| **Depth** | %PWT (pipe wall thickness percent) | GEV | 18.50 | 8.86 | 0.078 |
| **Length** | m | GEV | 0.095 | 0.195 | 0.753 |
| **Defect density** | Per ditch | Negative binomial | $n$ | $p$ | |
| | | | 0.208 | 0.21 | |

The goal of finding the probability distribution that represents the histogram of the length, depth of corrosion defect, and spatial density is to obtain sufficient information to use Bayes' rule. This rule needs information on the statistics of the variable studied as input; this expression is detailed in Equation (59):

$$P_o(\theta|\boldsymbol{X}) = \frac{L(\boldsymbol{X}|\theta)\pi(\theta)}{\int_\theta L(\boldsymbol{X}|\theta)\pi(\theta)d\theta} \tag{59}$$

where $\pi(\theta)$ is the prior distribution, $\boldsymbol{X}$ refers to the observed data, $L(\boldsymbol{X}|\theta)$ is the likelihood function to produce the strength of belief in parameter value $\theta$ (vector of parameters $\{\theta_i\}$) when the observed data $\boldsymbol{X}$ (vector of observed data $\{x_i\}$) are considered, and

$\int_\theta L(\boldsymbol{X}|\theta)\pi(\theta)d\theta$ is the marginal likelihood and represents the probability that the data follow the selected model under marginalization over all parameter values.

If the prior distribution of parameter $\theta_i$ is defined by the vector of parameters $\boldsymbol{\alpha_i}$, they are the hyperparameters of $\theta_i$.

If the number of ditches in the inspected pipeline is $\boldsymbol{n}_D$, the number of defects detected at the *kth* ditch is $\boldsymbol{n}_K$, and the observed defects in all these ditches sum up to $\boldsymbol{n}_T$, which can be written as follows:

$$L(\boldsymbol{Y}|\theta_D) = \prod_{1=1}^{\boldsymbol{n}_T} f_D\left(\boldsymbol{y_i}\big|\dot{\mu},\dot{\sigma},\zeta\right) \tag{60}$$

$$L(\boldsymbol{l}|\theta_l) = \prod_{1=1}^{\boldsymbol{n}_T} f_\wedge\left(\boldsymbol{l_i}\big|\dot{\mu},\dot{\sigma},\zeta\right) \tag{61}$$

$$L(\boldsymbol{N}|\theta_l) = \prod_{1=1}^{\boldsymbol{n}_T} f_{\mathrm{N}}(\boldsymbol{n_k}|n,p) \tag{62}$$

Equations (60) and (61) represent the PDFs of the depth and length of corrosion defects, respectively. In contrast, Equation (62) represents the PDF of the corrosion defect density. The prior distributions used in this study were based on previous experiences in other studies [14,15,71]. Normal and uniform priors were considered by the authors within a specific interval. The mean $(\mu)$ and variance $(\sigma^2)$ of these prior distributions are treated as hyperparameters of $\theta_i$; that is, $\boldsymbol{\alpha_i} = \{\mu, \sigma^2\}$ [17].

After the posterior distributions of $\theta$ were estimated, the probability of unobserved data can be feasibly predicted. If the data are expected to have a distribution $M$, then obtaining the predictive distribution $P_p(\hat{x}|\boldsymbol{X}, \boldsymbol{\alpha})$ of the unobserved data is likely. This predictive distribution can be estimated using the following mathematical expressions:

$$P_p(\hat{x}|\boldsymbol{X}, \boldsymbol{\alpha}) = \int_\theta M_x(x|\theta)P_o(\theta|\boldsymbol{X}, \boldsymbol{\alpha})d\theta \tag{63}$$

Therefore, the predictive distributions for the depth, length, and density of corrosion defects can be estimated using the following equations:

$$P_p(\hat{d}|\boldsymbol{Y}, \boldsymbol{\alpha_Y}) = \int_{\boldsymbol{\theta_D}} f_D(x|\boldsymbol{\theta_Y})P_o(\boldsymbol{\theta_D}|\boldsymbol{Y}, \boldsymbol{\alpha_D})d\boldsymbol{\theta_Y}; \tag{64}$$

$$P_p(\hat{l}|\boldsymbol{l}, \boldsymbol{\alpha_l}) = \int_{\boldsymbol{\theta_l}} f_D(x|\boldsymbol{\theta_l})P_o(\boldsymbol{\theta_l}|\boldsymbol{l}, \boldsymbol{\alpha_l})d\boldsymbol{\theta_l}; \tag{65}$$

$$P_p(\hat{n}|\boldsymbol{n}, \boldsymbol{\alpha_n}) = \int_{\boldsymbol{\theta_n}} f_D(x|\boldsymbol{\theta_n})P_o(\boldsymbol{\theta_n}|\boldsymbol{n}, \boldsymbol{\alpha_n})d\boldsymbol{\theta_n}; \tag{66}$$

The distributions obtained by these mathematical expressions constitute the final and most important result of the Bayesian methodology for the analysis of external corrosion defect data in the underground non-piggable pipelines studied by Valor [18] and Caleyo [17].

Considering the assumptions explained earlier on the independence of the variables of interest, the implementation of the proposed Bayesian methodology was based on the solution of three separate, moderately simple problems: one for depth (three parameters, six hyperparameters), another for length (idem), and the last for density (two parameters, four hyperparameters). To solve this problem, a numerical method called GRID was employed. This method is explained in the book titled *Doing Bayesian Data Analysis* [108].

This Bayesian methodology was applied to two pipeline systems to validate the process. Information is collected by an in-line inspection in a pipeline (PA) and inspections performed in a gathering pipeline (PB). Satisfactory results were obtained in both cases. These outcomes are shown in Figure 4 in [17]. In this figure, it can be observed that the predictive distribution obtained using BDA is close to the simulated empirical defects and all empirical (observed) defects. To assess the effectiveness of the results obtained using

BDA, they were compared with simulated empirical defects and all empirical (observed) defects using the Kolmogorov–Smirnov test and chi-squared test. The *p*-value of these tests is higher than 0.15 in all cases, meaning there is a high probability of accepting the hypothesis that the distribution obtained using BDA represents all corrosion defects present in the pipeline. Similar results were obtained when the corrosion defects presented in the gathering pipeline (PB) were analyzed using BDA; the distribution obtained was close to the simulated empirical defects and all empirical (observed) defects. In addition, the *p*-values for the Kolmogorov–Smirnov test and the chi-squared test are quite significant, indicating strong evidence towards the BDA methodology.

Another application of this Bayesian methodology is described by Valor et al. [18], where a pipeline failure rate is estimated by both the Markov model (Figure 4a in [17]) already described in this review [71] and a Bayesian updating (Figure 4b in [17]). The studied pipeline had a length, diameter, and a nominal wall thickness of 2.132 km, 20.32 cm, and 0.704 cm, respectively; the steel used in construction was API-5L-X52, and it was commissioned in 1985. The expected working pressure was 14 kg/cm$^2$. This pipeline was called A3L23 by the authors and underwent sampling inspection in 2009 within a total of 112 excavated ditches; 264 external corrosion defects were found. The failure indices of the A3L23 pipeline were also estimated using the corrosion defect depth, length, and density probability distributions obtained by sampling inspection; the final distribution for each characteristic was obtained using BDA, and the estimated parameters are shown in Table 4. In Figure 4a,b in [17], it can be observed that the estimations obtained using these two methods (Markov chain model and Bayesian updating) agree, with values in the same order of magnitude. This point indicates the suitability of the proposed reliability approach for a single, non-piggable pipeline using the defect size and density distributions estimated from previous information.

**Table 4.** Bayes-estimated parameters of the length and depth of corrosion defect and defect density in the studied pipeline [18].

| Defect Characteristic | Units | Fitted Distribution | Parameters | | |
|---|---|---|---|---|---|
| | | | $\mu$ | $\sigma$ | $\zeta$ |
| Depth | %PWT | GEV | 17.01 | 8.02 | 0.12 |
| Length | m | GEV | 0.078 | 0.086 | 1.11 |
| | | | $n$ | | $p$ |
| Density | Per ditch | Negative binomial | 0.35 | | 0.13 |

In general terms, Bayesian statistics is useful when the amount of available data about localized corrosion defect characteristics is not significant; therefore, this technique is advantageous for estimating reliability in non-piggable pipelines. Another advantage is that it is feasible to update the information as the number of inspections increases. On the other hand, a disadvantage of this method is that it is necessary to know the prior distributions of the variable to be studied to apply BDA.

A possible future outlook about Bayesian inference could consider the changes in mechanical properties that the pipeline undergoes because of the steel aging. Using BDA, it is possible to estimate the actual values of yield strength and UTS in a pipeline that has been in service for a long time.

## 8. The Future Challenge for the Application of Probability and Statistics in Corroded Oil and Gas Pipelines

After all the aforementioned applications of probability and statistics in corroded oil and gas pipelines, the future challenge can be described as follows:

I.      It would be interesting to study other electrochemical variables using statistical techniques. An example is the determination of whether the icorr or Epit of a steel sample in a corrosive environment exhibits stochastic behavior. This could help in determining the limits with a certain degree of confidence for some conditions where the corrosion rate could exist or the pitting potential could occur. Similarly, it would be necessary to correlate and model the pit initiation time using both electrochemical and statistical techniques for pipeline steels under aggressive environments. It is necessary to orient some studies using electrochemical impedance spectroscopy or electrochemical noise to obtain information that can help to model the pit initiation and the pit growth stochastically.

II.     The randomness of the pit initiation time is another parameter that should be studied and modeled. Velazquez et al. [14] determined different values of pit initiation time for each type of soil studied. In that study, the pit initiation time was found to be a regression parameter; therefore, it is considered a deterministic value. Nonetheless, determining whether this pit initiation time can also have characteristics of randomness and modeling this randomness would be applicable to estimating the remaining life of the pipeline because these corrosion defects do not initiate at the same time the pipeline makes contact with the soil.

III.    Further, finding a more accurate approach to determine the corrosion rate distribution in a pipeline using the information provided by consecutive in-line inspection (ILI) is worth mentioning. Many kilometers of oil and gas pipelines are inspected by ILI; however, it is difficult to determine the corrosion rate using this information because the technologies used in both inspections could be different, and the methods to calibrate the devices used differ; locating the same corrosion defects in two consecutive inspections can be a daunting task due to differences in the resolution of the devices used in each inspection, even if they are of the same technology.

IV.     Notwithstanding the evidence that the mechanical properties of pipeline steels change because of the aging of the material, as demonstrated by González-Arévalo et al. [23], there are no studies that use real-life information of aged and corroded pipelines to estimate their reliability. This may be because it is very difficult to monitor changes in the mechanical properties of an in-service pipeline. However, this estimation can help compute the failure probability with greater accuracy.

V.      BDA has been used in corroded pipelines to successfully estimate the remaining life and the failure probability. However, this statistical technique is not used for other factors related to pipeline deterioration, such as coating disbondment, cathodic protection, ECDA, conditions of fluid transmission, or even stray currents. This technique can be used to study all the phenomena involved in pipeline corrosion.

VI.     New approaches using machine learning techniques and probability concepts have been developed recently by Ossai [109]. These techniques will be widely used in years to come because they can include the independent variables that provoke the localized corrosion deterioration (temperature, chemical composition, fluid velocity, etc.) and represent the phenomenon's stochastic nature. One advantage of these approaches is that they can manage a vast amount of data with great flexibility.

## 9. Conclusions

Localized corrosion in oil and gas pipelines is a complicated phenomenon that involves several parameters such as the chemical composition of the steel, inclusion density, chemical composition of the environment that surrounds the pipe or of the fluid transmitted, coating type, stray currents, operation, and design of the cathodic protection system. Because of the numerous variables involved, the study of the phenomena as being of a random nature using probability and statistics has been proposed since the 1930s. From this study, the conclusions on the applications of probability and statistics in corroded pipelines are as follows:

- Pitting corrosion can be studied using the knowledge of probability and statistics, both in laboratory tests (electrochemical and immersion) and in-service pipelines. In electrochemical tests, the most studied variable is the corrosion potential ($E_{corr}$); however, there is a lack of deeper analysis of other characteristics, such as pitting or passive potential, to demonstrate their randomness. Conversely, several studies demonstrate by immersion tests the randomness of the pitting corrosion defect depth, not only in low-carbon steel (pipeline material) but also in other alloys. Usually, the deeper pitting corrosion defect depths measured after the immersion test can be fitted to a Gumbel distribution or GEVD with high confidence. For buried pipelines, the external depths of corrosion defect can also be fitted with high confidence in a GEVD.
- Regression analysis has been widely used to model the growth of localized corrosion defects with a sufficient confidence level. This type of statistical modeling has been used to predict the growth of external corrosion defects in buried pipelines and in solutions that simulate oilfield-produced water. Regression analysis is advantageous in that the physical and chemical characteristics of the environment that is in contact with the pipe can be incorporated into the model. Similarly, the initiation time of the corrosion defect can be included and a deterministic value can be obtained.
- Markov chain models have been successfully used to model the stochastic nature of localized corrosion defects in both immersion tests and oil and gas pipelines. In all cases, the Kolmogorov differential equations are the bases of the solutions in these models. These models correctly represent the shape, kurtosis, and skewness of the observed data histogram in pipeline inspections. In these models, it is also feasible to incorporate the chemical and physical characteristics of the environment in contact with the pipeline, meaning it is not only a purely mathematical model but also possible to establish a sound chemical and physical correlation between the characteristics of the corrosion defect and the properties of the environment.
- Both stochastic models and distributions fitted from observed data can be used to estimate the reliability of oil and gas pipelines. The accurate estimation of the depths of future corrosion defects drives a suitable pipeline reliability estimation, thus achieving better risk management because it is possible to channel resources at the most appropriate time.
- The Monte Carlo simulation approach was used to forecast the long-term distribution of the pitting corrosion rate and pipeline reliability estimation. This method has become quite popular because of the increasing computing power that allows complex simulations to be performed in a short time.
- Bayesian data analysis provides a useful tool for the estimation of the probability distributions of the corrosion defect depth, length, and density. This statistical method helps to estimate the conditions of the corrosion defect characteristics in oil and gas pipelines as long as there is prior information on probability distribution and some observed data in the inspections. BDA can be particularly useful in non-piggable pipelines because it is not feasible from the economic point of view to dig up an entire pipeline and carry out an inspection using a portable flaw detector. These pipelines are usually partially inspected; therefore, it is indispensable to infer the total damage of the structure. Using BDA, it is feasible to estimate the non-piggable pipeline reliability and in this way optimize resources in the maintenance plan.

**Author Contributions:** Conceptualization, J.C.V. and E.H.-S.; methodology, J.C.V. and G.T.; software, A.C.-T.; formal analysis, J.C.V. and S.C.-C.; investigation, J.C.V. and M.D.-C.; resources, J.C.V. and E.H.-S.; data curation, J.C.V. and E.H.-S.; writing—original draft preparation, J.C.V. and E.H.-S.; writing—review and editing, J.C.V. and E.H.-S.; project administration, J.C.V. All authors have read and agreed to the published version of the manuscript.

**Funding:** This work was supported by research Grant 20221070 of Instituto Politécnico Nacional in Mexico.

**Data Availability Statement:** The data that support the findings of this study are available from the corresponding author upon reasonable request.

**Acknowledgments:** The comments provided by the reviewers are deeply appreciated.

**Conflicts of Interest:** The authors declare no conflict of interest.

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
