# Peer review of "Probabilistic and Statistical Techniques to Study the Impact of Localized Corrosion Defects in Oil and Gas Pipelines: A Review"

_metals, doi:10.3390/met12040576_

Round 1

Reviewer 1 Report

Firstly, the reviewer would like to appreciate the authors for the clear structure of the manuscript and comprehensive material covered.  The authors thoroughly reviewed the probabilistic and statistical techniques to analyze the corrosion defects in oil and gas pipelines. The reviewer thinks the manuscript is comprehensive and worthy to publish.

Only some minor suggestions are put forth to the authors:

  • Since it is a review paper, the title should reflect that this article is a “review”, may be it is better to change the title to “review of …”
  • In the abstract, the author used “localize”, should it be “locate”?
  • May be the topic of the manuscript is not related to Covid-19 ?
  • The reviewer feels that the first 2 paragraph of the introduction is a little bit lengthy.
  • It seems that the 2nd section is not quite relevant to the main topic of the manuscript, it can be more concise.
  • It would be better if the author can specify the covariates and the dependent variable of the linear regression model in page 2, line 73.
  • It would be better if the authors can discuss the effect of all the factors in section 2 on the random process of pipe corrosion.
  • Is there any statistical assessment about the regression model?

Author Response

Reviewer 1

Firstly, the reviewer would like to appreciate the authors for the clear structure of the manuscript and comprehensive material covered. The authors thoroughly reviewed the probabilistic and statistical techniques to analyze the corrosion defects in oil and gas pipelines. The reviewer thinks the manuscript is comprehensive and worthy to publish.

Only some minor suggestions are put forth to the authors:

  • Since it is a review paper, the title should reflect that this article is a “review”, may be it is better to change the title to “review of …”
  1. The authors agreed to include this comment. In fact, the authors thanks for this comment.
  • In the abstract, the author used “localize”, should it be “locate”?
  1. This comment is applied.
  • May be the topic of the manuscript is not related to Covid-19 ?
  1. The authors agreed to remove the word related to Covid.
  • The reviewer feels that the first 2 paragraph of the introduction is a little bit lengthy.
  1. Because of the recent events about Ukraine-Rusia war, the following part was removed “ The Nord Stream 2 (system of natural gas pipelines running under the Baltic Sea from the Russian Federation directly to Germany with an estimated cost of 9500 million euros), which will transport 55 billion m3 per year with a length of 1230 km in a 1220 mm (48 inches) main pipeline diameter is another example. It is projected to be commissioned between 2021 and 2022. The purpose of this project is to provide energy security to Germany.”
  • It seems that the 2nd section is not quite relevant to the main topic of the manuscript, it can be more concise.
  1. Because of the electrochemical background of the corrosion phenomenon, the authors also considered that explaining the application of concepts of probability and statistics in electrochemical tests should be recalled. Likewise, the first application of probability of statistics began in electrochemical studies in corrosion phenomenon. The authors only include the central electrochemical studies that consider statistics as a tool. Therefore, the authors disagreed with this comment because we deemed explaining this process necessary.
  • It would be better if the author can specify the covariates and the dependent variable of the linear regression model in page 2, line 73.
  1. The model proposed by Romanoff that is indicated in page 2, line 73 is detailed in Section 3 in the manuscript. The authors added a footnote in line 73 to indicate that this model will be described later.
  • It would be better if the authors can discuss the effect of all the factors in section 2 on the random process of pipe corrosion.
  1. The authors agrees with this comment and it was decided to add the following sentence “As Aziz mentioned in the Reference [16], the randomness of the pitting corrosion phenomenon is due to the fact that the aforementioned factors and other minors (microscopic faults of the metal, weak spots in the oxide film or ions diffusion in the electrolyte) acting in a random fashion and produce erratic but predictable results.”
  • Is there any statistical assessment about the regression model?

Usually a regression model is assessed using the coefficient of determination (R2). The higher value of R2 the better performance of the selected model. This point is indicated and explained in line 437 and 475. The numerical values are not given because the observed data have a different origin.

Reviewer 2 Report

  1. Some of the references used for the paper seem to be not peer-reviewed ( e.g. references related to some news), and it is thus required for the authors to validate the information from such news by again citing other peer reviewed references .
  2. Most of the the figures taken from other references have low resolution. Necessary adaptations must be made to enhance the readability of these figures. If any changes are made to the original figures from the references, such adaptations need to be mentioned in  the figure caption as well.
  3. The number of figures can be reduced by combining similar figures.
  4. Too many equations are used (67 equations in total). 
  5. In fact, the manuscript looks like a report of the previous studies on the topics.  
  6. Though the manuscript cites recent literatures (2020, 2021), there are many many references that date back before 2000. This  makes the manuscript appear as based largely on a historical perspective on probabilistic/statistical models or techniques in materials corrosion studies. The state-of-the-art approaches and future outlook have not been highlighted less. Thus it is  recommended to make the manuscript more concise in relation to reporting of historical studies , and more elaborate on description of state-of-the-art models and methods in this area (e.g. data-driven and computational approaches in relation to the implementation statistical methods and probabilistic models for corrosion electrochemistry)

Author Response

Reviewer 2

  1. Some of the references used for the paper seem to be not peer-reviewed ( e.g. references related to some news), and it is thus required for the authors to validate the information from such news by again citing other peer reviewed references.

The information that the reviewer indicated as references related to some news was included to exemplify the importance of doing research in modeling corrosion defects growth in oil and gas pipelines. Nowadays, it seems to be fashionable to do research on renewable energies but it is important to indicate that oil and gas infrastructure will continue to be necessary. It is very common that research papers make reference to some “news” papers to indicate the situation of the topic in the industry. References [I,II,III] listed below illustrate this argument. Therefore, the authors consider that the information included in the manuscript help meet this goal.

[I] Martin Zerta, Patrick R. Schmidt, Christoph Stiller, Hubert Landinger; International Journal of Hydrogen Energy Volume 33, Issue 12, June 2008, Pages 3021-3025

[II] Lorcán Murray, The Electricity Journal Volume 32, Issue 6, July 2019, Pages 13-19

[III] Himani Negi, Carbohydrate Polymers Volume 266, 15 August 2021, 118125

  1. Most of the the figures taken from other references have low resolution. Necessary adaptations must be made to enhance the readability of these figures. If any changes are made to the original figures from the references, such adaptations need to be mentioned in the figure caption as well.

The authors agree regarding to enhance the readability of these figures. Some figures included in the manuscript were adapted from the originals. All figures that were changed are indicated in each figure caption. It is important to mention that the corresponding author of this manuscript (J.C. Velázquez) participated as author and coauthor in the papers where Figures 5, 6, 9, 10, 11 and 12 come from. For this reason, the authors have the data to modify certain plots.

  1. The number of figures can be reduced by combining similar figures.

The authors decided to remove Figures 13, 14 and 15 from the previous version of the manuscript. The text was updated in the manuscript's newest version to keep the explanation that the authors wanted to broadcast.

  1. Too many equations are used (67 equations in total).

Yes, it is true. To explain this topic and its application in reliability pipeline studies, it is necessary to explain all the equations presented. The authors wish to broadcast that there are many options to model the pipeline deterioration. To show these options is essential to include in the manuscript “the main” mathematical expression used in the different studies. Therefore, the authors insist on keeping the 67 equations in the manuscript.

  1. In fact, the manuscript looks like a report of the previous studies on the topics.

Unlike a report, which only describes the information, this manuscript includes the authors' comments for each method, the papers cited are analyzed and discussed, the advantages and disadvantages of each method are presented, and it is shown illustrations of the applications of each method. Likewise, the manuscript has a different structure that includes an introduction, future challenges and conclusions. The authors disagreed with this comment.

  1. Though the manuscript cites recent literatures (2020, 2021), there are many many references that date back before 2000. This makes the manuscript appear as based largely on a historical perspective on probabilistic/statistical models or techniques in materials corrosion studies. The state-of-the-art approaches and future outlook have not been highlighted less. Thus it is recommended to make the manuscript more concise in relation to reporting of historical studies, and more elaborate on description of state-of-the-art models and methods in this area (e.g. data-driven and computational approaches in relation to the implementation statistical methods and probabilistic models for corrosion electrochemistry)

The reviewer is right; the authors want to describe a historical perspective on statistical models used in pipeline corrosion studies in this manuscript. This was done to explain the origins of this topic because some concepts and mathematical expressions used in the first studies are still used in recent papers. The author also agreed in describing the future outlook of each method. This is explained and highlighted at the end of each section in the new version of the manuscript. On the other hand, the comment “make the manuscript more concise in relation to reporting of historical studies” is regarding the style of the manuscript. The authors insist in that this review is not a report. Finally, the authors are agreed in comment regarding the use of artificial intelligence methods in section 8, last issue. The use of artificial intelligence in pipeline corrosion is well recognized; however, the authors only want to include only statistical methods in this manuscript.

Reviewer 3 Report

The paper presented the literature review on probabilistic and statistical techniques to evaluate the impact of localized corrosion defects in oil and gas pipelines.  Reviewer proposed several comments related to the paper.

  1. Please change the title including the contents of the paper. i.e. review.
  2. Make more the necessary of the review study clear in the induction.
  3. Provide the advantage, disadvantage and limitation of the each method.

Author Response

Reviewer 3

The paper presented the literature review on probabilistic and statistical techniques to evaluate the impact of localized corrosion defects in oil and gas pipelines. Reviewer proposed several comments related to the paper.

  1. Please change the title including the contents of the paper. i.e. review.

The authors are totally agreed. The title of the manuscript was updated.

  1. Make more the necessary of the review study clear in the induction.

The authors agreed on this comment. For this reason, the following paragraph was added in the introduction section: “Several pipeline corrosion studies have been conducted using probability and statistics as a tool to manage structural integrity, thereby reducing the risk of leaks and ruptures. In this context, the present review summarizes the most used and recognized statistical techniques applied in oil and gas pipelines and some exemplifications of these. In addition, the basics for these statistical techniques are explained in each section with the purpose to simplify the information search for future specialists.”

  1. Provide the advantage, disadvantage and limitation of the each method.

The authors coincide with the comment. At the end of each section is added a paragraph where is discussed the advantage and disadvantage of each method. These paragraphs are highlighted to facilitate the location.

Reviewer 4 Report

The paper deals with the localized corrosion defects in oil and gas pipelines reviewing the status of the related research work and is considered authors’ own work based on huge number of reference papers which may contribute to summarize the related works. Besides, the paper is recommended to be revised carefully. The followings are the comments for authors.

  1. The appearance of some equations seems wrong. For example, equation (2) seems wrong. Equation (4); the appearance of H2O seems wrong. Not only of these, please be check the appearance all through the manuscript.
  2. The reference should be checked again carefully. For example, Line 509; Provan and Rodrigues [72] seems wrong. The reviewer won’t be able to check the whole.
  3. Another problem of the paper is that some of the referenced papers are sometimes written such as “Bazan and Beck 2013 (Line 765)” not using reference number. Too many such parts are remained and recommended to be modified appropriately.
  4. In some parts of the paper, one unit is written as “%pwt” which is uncertain for the reviewer. Please ignore if it is well known one.

Author Response

The paper deals with the localized corrosion defects in oil and gas pipelines reviewing the status of the related research work and is considered authors’ own work based on huge number of reference papers which may contribute to summarize the related works. Besides, the paper is recommended to be revised carefully. The followings are the comments for authors.

  1. The appearance of some equations seems wrong. For example, equation (2) seems wrong. Equation (4); the appearance of H2O seems wrong. Not only of these, please be check the appearance all through the manuscript.

This reviewer is correct. The authors have verified all chemical equations and all chemical notations to avoid more typo mistakes in the manuscript.

  1. The reference should be checked again carefully. For example, Line 509; Provan and Rodrigues [72] seems wrong. The reviewer won’t be able to check the whole.

The reviewer is ok. In this case, the Reference [72] is the continuation of Reference [71]. Reference [71] is part 1 and [72] is part 2. However, the authors' list order is switched in the second part. So, Reference [71] is cited as Provan and Rodriguez, but Reference [72] is cited as Rodriguez and Provan. The corrections were carried out and also the reference list was verified. Thanks for the comment.

  1. Another problem of the paper is that some of the referenced papers are sometimes written such as “Bazan and Beck 2013 (Line 765)” not using reference number. Too many such parts are remained and recommended to be modified appropriately.

The authors thank this comment and proceed to the application. Some references were not updated to the journal format but now were verified and modified in some of them.

  1. In some parts of the paper, one unit is written as “%pwt” which is uncertain for the reviewer. Please ignore if it is well known one.

The term %pwt means pipe wall thickness percent. It is a term commonly used in pipeline inspections. Reference [22] exemplifies the used of this term. To clarify this issue, a footnote is added.

Round 2

Reviewer 2 Report

The revised manuscript ignores my previous comments

  1. Use of references that are not peer-reviewed: More than 5 references are not peer-reviewed, and sometimes such references are opinions rather than validated facts.  Particularly, some information on  some references seem to be dynamically changing regardless of the opinion expressed in the references. Peer-reviewed journals have no space for unproven claims. Only if the opinions put forward in the references are validated with facts (mathematically proven or peer-reviewed contents), then the present manuscript can be considered for publication. 

By providing the examples of other articles that also use news content does not validate the opinions mentioned in the references of this manuscript. 

2. The revised manuscript is in fact not different than the original manuscript .  All of the comments (except the comment on the conciseness of the graphical presentation) have been ignored . 

In summary, the manuscript is not publishable in its present form.

Author Response

The authors would like to thank you again for your careful and thorough reading of our manuscript. All of your comments are valuable and helpful for improving our paper. The revision has been made according to your suggestions and highlighted in the text.

Reviewer’s comments:

  1. Use of references that are not peer-reviewed: More than 5 references are not peer-reviewed, and sometimes such references are opinions rather than validated facts.  Particularly, some information on some references seem to be dynamically changing regardless of the opinion expressed in the references. Peer-reviewed journals have no space for unproven claims. Only if the opinions put forward in the references are validated with facts (mathematically proven or peer-reviewed contents), then the present manuscript can be considered for publication.

By providing the examples of other articles that also use news content does not validate the opinions mentioned in the references of this manuscript.

  1. All references that the reviewer called “references related to some news” were removed. However, the authors decided to keep using references where some statistics information were considered (e.g. British petroleum outlook 2020, CIA world factbook or EGIG 11th Report of the European Gas Pipeline Incident Data Group). The references list was updated because of this fact. In addition, the following sentences were included in the first paragraph of the manuscript:

“However, fossil fuels are indispensable for economic growth in many countries. Some studies have indicated that the oil demand will probably peak in the second part of the next decade (2030-2040) but will be highly demanded in developing countries for more time”

  1. The revised manuscript is in fact not different than the original manuscript .  All of the comments (except the comment on the conciseness of the graphical presentation) have been ignored . 
  2. The authors disagreed with this comment. In this new version of the manuscript, only the changes derived from the reviewer’s suggestions are highlighted to facilitate their location. Recalling the previous comments, the authors clarify the following points:

1.- The authors enhanced the readability of several figures, as the reviewer suggested.

2.- The authors updated the figures’ captions following the reviewer’s recommendation.

3.- Figures 13, 14 and 15 were removed with the purpose to reduce the number of figures as the reviewer suggested

Point 4 is related to the number of equations listed in the paper. The authors reject modifying the list because all these equations are considered necessary to explain the models used in corroded pipelines. Authors do not include deductions or redundant mathematical expressions. The authors only want to condense the “main equations” related to this topic in one document.

In the last issue listed in the previous comments done by the reviewer, it is suggested to include some studies that used artificial intelligence as a tool to estimate the pipeline’s remaining life. However, this is not considered a purely statistic/ probabilistic technique. But, the authors included a paragraph in section 8 “Future challenges” because it is feasible to build a kind of “hybrid model” that uses concepts and mathematical expressions of statistics with the flexibility of machine learning tools. In addition, at the end of each section, one paragraph is added to explain the future outlook of each method.

The authors applied several comments gratefully done by the reviewer. Unfortunately, some comments would modify the original purpose that the authors want to broadcast that is to summarize the main models found in the literature and condense them in a paper to guide future researchers. 

Reviewer 3 Report

Reviewer's comments was well addressed. The paper can be published.

Author Response

Thank you for your comments and suggestions

Round 3

Reviewer 2 Report

The updated and revised manuscript now addresses the  comment 1 made for the previous version of the manuscript. I recommend for the acceptance of the manuscript.